

**Urban increments of gaseous and aerosol pollutants and**
**their sources using mobile aerosol mass spectrometry**
**measurements**
**M. Elser[1], C. Bozzetti[1], I. El-Haddad[1], M. Maasikmets[2], E. Teinemaa[2], R. Richter[1],**
**R. Wolf[1], J.G. Slowik[1], U. Baltensperger[1] and A.S.H. Prévôt[1]**
[1]{Laboratory of Atmospheric Chemistry, Paul Scherrer Institute, 5232, Villigen PSI,
Switzerland}
[2]{Estonian Environmental Research Centre, 10617, Tallinn, Estonia}
Correspondence to: I. El-Haddad (imad.el-haddad@psi.ch) and A. S. H. Prévôt
(andre.prevot@psi.ch)
**Abstract**
Air pollution is one of the main environmental concerns in urban areas, where anthropogenic
emissions strongly affect air quality. This work presents the first spatially-resolved detailed
characterization of the $PM_{2.5}$ in two major Estonian cities (Tallinn and Tartu), using mobile
measurements. In both cities, the non-refractory (NR)-$PM_{2.5}$ was characterized by a high-
resolution time-of-flight aerosol mass spectrometer (HR-ToF-AMS) using a recently
developed lens which increases the transmission of super-micron particles. Equivalent black
carbon (eBC) and several trace gases including carbon monoxide (CO), carbon dioxide ($CO_2$)
and methane ($CH_4$) were also measured. The chemical composition of the $PM_{2.5}$ was found to
be very similar in the two cities. Organic aerosol (OA) constituted the largest fraction,
explaining on average about 52 to 60 % of the $PM_{2.5}$ mass. Four sources of OA were
identified using positive matrix factorization (PMF): hydrocarbon-like OA (HOA, from
traffic emissions), biomass burning OA (BBOA, from biomass combustion), residential
influenced OA (RIOA, probably mostly from cooking processes with possible contributions
from waste and coal burning) and oxygenated OA (OOA, related to secondary aerosol
formation). OOA was the major OA source during night-time, explaining on average half of
the OA mass, while during day-time mobile measurements the OA was affected by point
sources and dominated by the primary fraction. A strong increase in the secondary organic
and inorganic components was observed during periods with transport of air masses from





polluted continental areas, while the primary local emissions accumulated during periods
with temperature inversions. Mobile measurements offered the identification of different
source regions within the urban areas and an accurate calculation of the urban increments.
HOA, eBC, $CO_2$ and CO showed stronger enhancements on busy roads during the morning
and evening traffic rush hours; BBOA had its maximum enhancement in the residential areas
during the evening hours and RIOA was enhanced in both the city center (emissions from
restaurants) and in the residential areas (emissions from residential cooking). In contrast,
secondary components (OOA, $SO_4$, $NO_3$, $NH_4$, and Cl) had very homogeneous distributions
in time and space. We were able to determine a total $PM_{2.5}$ urban increment in Tartu of 6.0
$\mu g\ m^{-3}$ over a regional background concentration of 4.0 $\mu g\ m^{-3}$ (i.e., a factor of 2.5 increase).
Traffic exhaust emissions were identified as the most important source of this increase, with
eBC and HOA explaining on average 53.3 and 20.5 % of the total increment, respectively.

## 1   Introduction

Atmospheric particulate matter (PM) plays a central role in many environmental processes
through its influence on climate (radiative forcing; Myhre et al., 2013), the hydrological cycle
(Ramanathan, et al., 2001) and its adverse effects on health (Pope and Dockery, 2006).
Recently, major attention has been devoted to the study of the $PM_{2.5}$ fraction (particulate
matter with an aerodynamic equivalent diameter $d_{aero} \leq 2.5\ \mu m$), which has been linked to
increased lung cancer rates (Hu and Jiang, 2014), acute bronchitis and asthma (Gao et al.,
2015), and mortality (Dockery et al 1993; Laden et al., 2006). Atmospheric particles can be
classified as primary or secondary aerosols according to their formation processes. Primary
particles are directly emitted, while secondary aerosols are formed from gas-phase precursors
following chemical transformation in the atmosphere. Aerosols can be further classified in
terms of their emission sources as natural sources (e.g. volcanic eruptions, wildfires, sea salt,
dust or biogenic emissions from plants) or anthropogenic sources (mostly from combustion
processes, e.g. traffic and residential wood combustion).
Due to enhanced contributions of anthropogenic sources, air quality is commonly lower in
urban areas compared to rural or suburban locations (Putaud et al., 2004). In Europe, annual
average $PM_{2.5}$ mass concentrations in urban areas commonly vary between a few $\mu g\ m^{-3}$ up to
35 $\mu g\ m^{-3}$ (Putaud et al., 2010). The predominance of specific aerosol sources (e.g.
residential, traffic, industry) or the implementation of new technologies (e.g. car fleet, heating





systems, etc.) may strongly influence the levels and physicochemical characteristics of the
pollutants in these locations. Moreover, certain orographic features and stagnant
meteorological conditions may induce the accumulation of local pollutants (Putaud et al.,
2004; Carbone et al., 2010; Squizzato et al. 2012). Likewise, long-range transport of
continental air masses has been shown to influence the PM in different urban areas in Europe
(Niemi et al., 2009; Baker, 2010; Salvador et al., 2013; Beekmann et al., 2015; Di Gilioa et
al., 2015; Ulevicius et al., 2015). While the PM levels and physicochemical properties of the
particles are well characterized in Western Europe, data are scarce in Eastern European cities,
especially in the Baltic region, hindering air quality assessment and quantification of the main
aerosol sources.
In contrast to conventional stationary measurements, mobile measurements (including
zeppelin, aircraft and ground measurements) are suited for pollutant mapping, chasing of
mobile sources or measurements in emission plumes, etc. Ground-based measurements by
mobile platforms have been successfully performed in the last years to measure particles and
trace gases from real-world traffic emissions (Pirjola et al., 2004, 2006 and 2012; Kwak et al.,
2014; Kyung Hwan et al., 2015) and from wood burning emissions (Pirjola et al., 2015).
More recently, aerosol mass spectrometers (AMS) have been deployed in mobile laboratories
in order to determine the physical and chemical properties of submicron aerosols ($PM_1$,
particulate matter with aerodynamic equivalent diameter $d_{aero} \leq 1$ μm) in urban environments
like Zurich (Mohr et al., 2011), Paris (Von der Weiden-Reinmueller et al., 2014a and 2014b)
or Barcelona (Mohr et al., 2015). Moreover, a newly developed inlet for the AMS has been
used to measure the chemical composition of the non-refractory (NR)-$PM_{2.5}$ fraction in
Bologna (Wolf et al., 2015).
In this work we present the first detailed in-situ mass spectrometric measurements of air
pollutants in the two biggest cities in Estonia (Tallinn and Tartu). The measurements were
performed using the Paul Scherrer Institute (PSI) mobile laboratory (Bukowiecki et al., 2002;
Mohr et al., 2011; Wolf et al., 2015). The use of a high-resolution time-of-flight aerosol mass
spectrometer (HR-ToF-AMS) with a novel $PM_{2.5}$ lens allowed for a detailed characterization
of the NR-$PM_{2.5}$ fraction in the measurement areas. The spatial distributions of the sources of
organic aerosols (OA), inorganic aerosols (nitrate ($NO_3$), sulfate ($SO_4$), ammonium ($NH_4$),
and chloride (Cl)), equivalent black carbon (eBC) and some of the major gas-phase
components (carbon monoxide (CO), carbon dioxide ($CO_2$) and methane ($CH_4$)) were
determined in the urban areas. Such analyses allowed for the calculation of regional



background and urban concentrations of the different gas- and particle-phase components and
provided direct insights into the spatial resolution of local emissions and their impact on the
air quality in different city areas. Long-range transport of pollutants and accumulation events
as well as their effect on the particle- and gas-phase mass concentrations will also be
discussed.

## 2 Methodologies

### 2.1 Measurement campaign

The measurements were performed in the two biggest cities in Estonia. Tallinn, the capital
and the largest city of Estonia, has a population of 413,000 inhabitants (Statistical Database,
2015) and occupies an area of 158.3 km$^2$. Located on the northern coast of the country,
Tallinn has some of the biggest ports in the Baltic Sea. Among them, the old city harbor is
one of the busiest passenger harbors in the region. Tartu, with 38.8 km$^2$ and more than 97,000
inhabitants in 2015 (Statistical Database, 2015), is the second largest city in Estonia. The city
is situated in the center of southern Estonia, in the post-glacial valley of the Emajõgi River,
which influences the local meteorological conditions and favors the accumulation of local
pollutants under frequent temperature inversions. Previous studies identified traffic emissions
and residential heating as the major sources of air pollution in these two cities (Urb et al.,
2005; Orru et al., 2011). An older vehicle fleet, the limited network capacity of the city
streets (which generates congestions during rush hours) and the extensive use of studded
tires, have been reported to strongly enhance the effect of the traffic emissions in the city
center and major roads. Residential heating includes extensive use of inefficient wood and
coal stoves with low stacks in both cities. In this regard, a detailed modeling study performed
in Tallinn and Tartu (Orru et al., 2011) revealed that the city centers and the neighborhoods
with local heating are exposed to much higher average PM$_{2.5}$ concentrations compared to
other areas of the cities.
The measurements took place from 10 to 17 March 2014 in Tartu and from 25 March to 1
April 2014 in Tallinn. The GPS trace of the driving routes in the two cities is shown in Fig. 1.
The paths were chosen in order to cover heavily trafficked roads, residential areas where
different heating systems are used (wood/coal burning, central heating or mixed) and
background sites with little local emissions. In Tallinn, streets close to the old town harbor
were also included in the route. To obtain statistically significant spatial distributions of the



major pollutants, 25 loops were performed at each location throughout the measurement
periods at different times of the day. The average loop duration was about 72 minutes in
Tartu and 112 minutes in Tallinn. Stationary measurements were typically performed
overnight at a gasoline station in Tartu (influenced by city center and residential emissions)
and at the Estonian Environmental Research Centre (EERC) in Tallinn (a background site).
Meteorological data were recorded in a meteorological-tower in Külitse (around 10 km south-
east from Tartu) and in the Tallinn-Zoo meteorological station.

## 2.2   Mobile laboratory set-up

A schematic of the instrumental set-up in the PSI mobile platform is shown in Fig. S1. The
main inlet of the mobile platform was kept at a constant flow of ~11 m sec$^{-1}$ for isokinetic
sampling during driving conditions, assuming an average velocity in the city of ~ 40 km h$^{-1}$.
Two different inlet lines connected the main inlet to the aerosol and gas-phase
instrumentation. The deployed instruments, measured parameters and their time resolution
are listed in Table 1. All parameters were determined with high time resolution (between 1
and 25 seconds), critical for the identification of source regions using a mobile platform.
An HR-ToF-AMS (Aerodyne Research Inc.) was deployed to measure the chemical
composition of the NR-PM$_{2.5}$ aerosol, including NO$_3$, SO$_4$, NH$_4$, Cl, and OA. For this work
the AMS was equipped with a recently developed aerodynamic lens which extends the
measured particle size to the PM$_{2.5}$ fraction (in contrast to the conventional PM$_1$ lens). The
PM$_{2.5}$ lens efficiently transmits particles between 80 nm and up to at least 3 µm and has been
well characterized by Williams et al. (2013) and tested in previous chamber and ambient
studies (Wolf et al., 2015; Elser et al., 2015). The operating principle of the instrument can be
found elsewhere (DeCarlo et al., 2006). A nafion drier (Perma Pure MD-110) was set before
the AMS inlet in order to dry the ambient particles and reduce uncertainties in the bounce-
related collection efficiency (CE$_b$) and possible transmission losses of large particles at high
relative humidity (RH).
A 7-wavelength Aethalometer (Magee Scientific, model AE33) was used to measure the
aerosol light absorption and to determine the equivalent black carbon (eBC) concentrations.
The measurement at 7 different wavelengths (370, 470, 520, 590, 660, 880 and 950 nm)
covers the range between ultraviolet and infrared and allows for the source apportionment of
different eBC fractions (Sandradewi et al., 2008; Zotter et al., in prep). Moreover, the dual





spot measurement method corrects automatically for the loading effect and provides a real-
time calculation of the loading compensation parameter (Drinovec et al., 2015).
The concentrations of trace gases, including CO, $CO_2$ and $CH_4$ were measured by two
different analyzers (Picarro-G2301 and Licor-6262). In addition, some important parameters
for mobile measurements (GPS, temperature, relative humidity and solar radiation) were also
measured continuously.
**2.3   AMS data analysis**
AMS data were analyzed in Igor Pro 6.3 (WaveMetrics) using the standard ToF-AMS Data
Analysis toolkit (SQUIRREL version 1.53G and PIKA version 1.12G). Based on standard
$NH_4NO_3$ calibrations, the ionization efficiency (IE, defined as ions detected per molecules
vaporized) was determined to be $5.08 \cdot 10^{-8}$ (average of five calibrations during the full
measurement period). Standard relative ionization efficiencies (RIE) were used for nitrate,
chloride, and organics (RIE = 1.1, 1.3, and 1.4, respectively) and experimentally determined
for sulfate and ammonium (RIE = 1.11 and 4.29, respectively). A composition dependent
collection efficiency (CE) algorithm by Middlebrook et al. (2012) was used to calculate the
ambient mass concentrations.
**2.4   Source apportionment techniques**
2.4.1  OA source apportionment
To identify and quantify the major sources of OA in the different measurement areas, positive
matrix factorization (PMF; Paatero and Tapper (1994)) was applied to the highly time
resolved AMS data (see Table 1). The analysis were performed using the multilinear engine
tool (ME-2; Paatero, 1997) implemented in the Source Finder interface (SoFi; Canonaco et
al., 2013) coded in Igor Wavemetrics.
PMF is a bilinear unmixing algorithm which, as defined in Eq. (1), allows representing a two-
dimensional matrix of measured data ($\mathbf{X}$) as a linear combination of a given number of static
factors profiles ($\mathbf{F}$) and their corresponding time series ($\mathbf{G}$). The matrix $\mathbf{E}$ in Eq. (1) contains
the model residuals. The model uses a least squares approach to iteratively minimize the
object function Q described in Eq. (2):
$$\mathbf{X} = \mathbf{GF} + \mathbf{E} \tag{1}$$





$$Q = \sum_{i=1}^{m} \sum_{j=1}^{n} \left( \frac{e_{ij}}{\sigma_{ij}} \right)^2 \qquad (2)$$

where $e_{ij}$ are the elements from the error matrix (E) and $\sigma_{ij}$ are the respective uncertainties of
**X**.
In our case, the model input consists of a data and error matrix of OA mass spectra, where the
rows represent the time series (62665 points, with steps of 25 seconds) and the columns
contain the ions fitted in high resolution (292 ions). The organic mass obtained from the high
resolution fits (up to $m/z$ 115) agrees with the mass from the unit mass resolution fits (up to
$m/z$ 737) within ± 5 %. The initial error values were calculated with the HR-AMS data
analysis software PIKA. A minimum error corresponding to the measurement of a single ion
was applied (Ulbrich et al., 2009).  All variables with signal-to-noise ratio (SNR) lower than
0.2 were removed and the variables with SNR between 0.2 and 2 were down-weighted by
increasing their error by a factor of 3 (Paatero and Hopke, 2003). Moreover, all variables
directly calculated from the $CO_2^+$ in the organic fragmentation table (i.e. $O^+$, $HO^+$, $H_2O^+$ and
$CO^+$) (Allan et al., 2004) were excluded for the PMF analysis to appropriately weight the
variability of $CO_2^+$ in the algorithm and were reinserted post-analysis.
The possibility of local minima in the solution space and the uncertainty of the PMF solution
were investigated by means of bootstrap analysis. This statistical method is based on the
creation of replicate datasets perturbing the original data by resampling. In each replicate,
some randomly chosen rows of the original matrix are present several times, while other rows
do not occur at all (Paatero et al., 2014), such that the dimension of the data matrix is kept
constant. This resulted in about 64 % of the original points being used in each replicate. PMF
was applied to 100 different replicates and the variations among these results were used to
estimate the uncertainty of the initial PMF solution. Note that as each bootstrap run is started
from a different initialization point and hence this methodology includes the investigation of
the classic seed variability. All convergent solutions were found to be consistent, an
indication of the robustness of the chosen solution.
The results presented in this section were obtained by merging the measurements from the
two measurement locations, as no major changes were observed if the source apportionment
was performed for the individual cities.



## 2.4.2  eBC source apportionment

The Aethalometer measurements can be used to separate eBC from wood burning ($eBC_{wb}$) and from traffic ($eBC_{tr}$), by taking advantage of the spectral dependence of absorption, as described by the Ångström exponent (Ångström, 1929). This method is described in detail in Sandradewi et al. (2008) and has been successfully applied at many locations across Europe. For a proper separation of the eBC fractions, the Aethalometer data was averaged to 30 minutes in order to increase the signal to noise. Thus, the obtained fractions $eBC_{wb}$ and $eBC_{tr}$ could only be used for the correlations with the external tracers, but their spatial distributions couldn't be explored. The absorption Ångström exponent was calculated using the absorption measured at 470 and 950 nm and Ångström exponents of 0.9 and 1.7 were used for traffic and wood burning, respectively, following the suggestions in Zotter et al. (In prep.).

## 3    Results and discussion

### 3.1    Pollutant concentrations and temporal variability

The temporal variation of all measured gas- and particle-phase components is shown in Fig. 2a. The type of measurement is indicated by different background colors (transparent for stationary measurements and orange for mobile measurements). The measurement period included three distinct meteorological periods of transport of polluted air masses and accumulation of local emissions. These periods are referred to as special events (indicated by a red frame) and will be treated separately and discussed in detail in Section 3.4. While the AMS and Aethalometer were running almost continuously during the entire measurement period, there is a small gap in the $CO_2$, CO and $CH_4$ data due to an instrument malfunction. Over the full measurement period, the average mass concentration of $PM_{2.5}$ (NR-$PM_{2.5}$ plus eBC) was 12.3 µg m$^{-3}$. In the gas-phase, average concentrations of 414.1 ppm of $CO_2$, 0.24 ppm of CO and 1.92 ppm of $CH_4$ were measured. In contrast to these relatively low average values, extremely high concentrations were often recorded during the mobile measurements due to local emissions from point sources (around 50 spikes with $PM_{2.5}$ mass concentration exceeding 100 µg m$^{-3}$). Such intermittent pollution plumes (expected in some areas in a city) cannot be detected from stationary measurements at an urban background site, but enhance negative health impacts. As shown in Fig. 2b, neglecting the periods defined as special events, the $PM_{2.5}$ average concentrations and relative contributions of the particle phase



species were very similar at the two locations. If we compare day-time (07:00 to 19:00, local
time (LT)) and night-time (19:00 to 07:00, LT) measurements, in both cities the average
$PM_{2.5}$ was higher during the day (11.0 µg m$^{-3}$ in Tartu and 11.6 µg m$^{-3}$ in Tallinn) than during
the night (6.5 µg m$^{-3}$ in Tartu and 7.1 µg m$^{-3}$ in Tallinn), despite the development of the
boundary layer and increased dilution during day-time. OA constituted in all cases the largest
mass fraction, explaining on average 52.2 and 54.3 % of the $PM_{2.5}$ mass in Tartu (during
night- and day-time, respectively) and 55.2 and 60.1 % in Tallinn (during day- and night-
time, respectively). Primary emissions of eBC contributed on average 20.4 % and 33.7 % in
Tartu (during night-time and day-time, respectively), and 13.4 and 26.9 % in Tallinn (during
night-time and day-time, respectively), constituting a substantially higher fraction than at
other European locations (Putaud et al., 2010). The remaining mass, 12 to 28 %, was related
to secondary inorganic species, mostly ammonium sulfate and nitrate. These species were
found to be neutralized within the uncertainties (ratio of $NH_4$ expected from an ion balance to
$NH_4$ measured of 1.05, with $R^2$=0.95). During night-time a decrease in the relative
contribution of eBC was observed in favor of an enhanced contribution of the inorganic
species.

## 3.2  Sources of OA

To properly represent the temporal variations of the OA, four factors were required:
hydrocarbon-like OA (HOA), biomass burning OA (BBOA), residential influenced OA
(RIOA) and oxygenated OA (OOA). The mass spectra of these factors are reported in Fig. 3.
HOA is a primary source related to traffic emissions and its mass spectrum is characterized
by the presence of alkyl fragment signatures (Ng et al., 2011), with prominent contributions
of non-oxygenated species at $m/z$ 43 ($C_3H_7^+$), $m/z$ 55 ($C_4H_7^+$) and $m/z$ 57 ($C_4H_9^+$). As shown
in Fig. S2, a fairly good correlation is found between HOA and $eBC_{tr}$ ($R^2 = 0.4$). Moreover,
the ratio of HOA to $eBC_{tr}$ was 0.5, which is in good agreement with previous European
studies (El Haddad et al., 2013 and references therein). BBOA is associated with domestic
heating and/or agricultural biomass burning activities, and shows characteristic high
contributions of the oxygenated hydrocarbons at $m/z$ 60 ($C_2H_4O_2^+$) and $m/z$ 73 ($C_3H_5O_2^+$),
which are known fragments from anhydrous sugars (Alfarra et al., 2007). BBOA correlates
fairly well with $eBC_{wb}$ ($R^2 = 0.4$), and the ratio of BBOA to $eBC_{wb}$ was 4.0 (Fig. S2), which
within the method uncertainties is consistent with previously reported values (Crippa et al.,
2013). The ratio BBOA to $eBC_{wb}$ was found to be very sensitive to the chosen Ångström





exponent for traffic, and it increased to 4.8 if a slightly higher Ångström exponent (i.e. 1.0
instead of 0.9) was considered for traffic. RIOA is a hydrocarbon-rich factor that was
required for a reasonable explanation of the variability in the data. Due to its increase in the
residential areas, this factor was associated with residential emissions. Given its strong
correlation ($R^2 = 0.9$) with cooking markers such as the fragment ion $C_6H_{10}O^+$ at $m/z$ 98 (Sun
et al., 2011; Crippa et al., 2013), we expect that a great part of this factor is related to cooking
emissions (see Fig. S2). Moreover, as in previously reported cooking spectra (Mohr et al.,
2012), the RIOA mass spectrum shows a higher $m/z$ 55 to $m/z$ 57 ratio than HOA. However,
in the absence of diurnal trends due to the driving conditions, the separation of cooking
emissions from other residential emissions (such as domestic coal and waste burning) was not
possible. OOA is associated with aged emissions and secondary organic aerosol formation,
and its profile is characterized by a very high $m/z$ 44 ($CO_2^+$). In general, OOA increases
simultaneously with the secondary species (especially $NO_3$), but the ratio among these
components changes during special events (Fig. S2). If the number of factors is decreased,
the RIOA factor is not resolved and the OOA time-series becomes contaminated by local
spikes, which is unexpected for a regional component (see Fig. S3 and S4). In contrast, if a
five-factor solution is considered an additional highly oxygenated factor is obtained
("unknown" factor in Fig. S3 and S4). The mass spectrum of this additional factor resembles
a low-volatility OOA (LV-OOA), as resolved in many previous works (Jimenez et al., 2009),
but its time series exhibits the typical characteristics of the primary factors, i.e. strong
increases in emission areas. Therefore, this further increase in the number of factors doesn't
seem to improve the interpretation of the data, as the new factor cannot be explicitly
associated to distinct sources or processes. Accordingly, a four-factor solution was
considered as optimal and is utilized below.
Figure 4a represents the time series of the absolute mass (top panel) and relative contributions
(bottom panel) of the retrieved OA sources for the two measurement locations. The
variability of these time series over 100 bootstrap runs was relatively low, as shown in Fig.
S5. In both cities, the three primary sources (HOA, BBOA and RIOA) exhibit a very spiky
temporal behavior, while the secondary OOA is characterized by a relatively smooth time
series. Figure 4b reports the averaged total OA mass and relative contributions of the OA
sources during the measurements in Tartu (top panel) and Tallinn (bottom panel). The
reported errors (which correspond to the standard deviation among 100 bootstrap runs) are an
indication of the high stability of the solution. Overall, the relative errors vary between 3 %



and 7 %, except for the RIOA, which shows slightly higher variability during night-time
(relative error of 11 % in Tartu and 13 % in Tallinn). Similarly to the total $PM_{2.5}$ mass and as
reported in Fig. 4b, neglecting the special events, a strong daily cycle can be observed in the
total OA mass, with higher concentrations during day-time (6.0 µg m$^{-3}$ and 6.3 µg m$^{-3}$ in
Tartu and Tallinn, respectively) than during night-time (3.4 µg m$^{-3}$ and 4.2 µg m$^{-3}$ in Tartu
and Tallinn, respectively). This difference is mostly driven by the increase of primary aerosol
emissions (HOA, BBOA and RIOA) during the day. This structure is observed independently
of the nature of the measurements (stationary or mobile), indicating that except for the
periods where emissions from point sources are sampled, the OA concentrations and sources
are rather homogeneous across the sampling area. In terms of relative contribution, OOA is
dominant during night-time, explaining on average between 42 and 44 % of the OA mass in
Tartu and Tallinn, respectively. HOA and RIOA relative contributions to the total OA are
higher during day-time (the relative contribution of HOA increases from about 20 to 32% in
Tartu and from 11 to 27% in Tallinn; the relative contribution of RIOA increases from 20 to
27 % in Tartu and from 20 to 22 % in Tallinn). BBOA shows similar relative contributions
for day- and night-time in Tartu (explaining about 17 % of the OA mass), and slightly lower
during the day-time in Tallinn (20 % during day-time and 25 % at night-time).
## 3.3  Spatial distributions, regional background and urban increments
The average spatial distributions of the four OA sources, $SO_4$, $NO_3$, eBC, $CO_2$ and CO are
represented in Fig. 5 and 6 for Tartu and Tallinn, respectively. The spatial distributions of the
additionally measured gas and particle components are reported in Fig. S6 and S7. All loops
for which all the instruments were running (except $CO_2$, CO and $CH_4$ in Tallinn) were
averaged on a grid with grid cells of 250 m$^2$. In order to get comparable distributions from
different days of measurements, the 5$^{th}$ percentile (P05) of was subtracted from each single
loop for all components. The subtraction of P05 was found to be optimal to decrease the
variability among different loops enough to make them comparable. However, as it will be
discussed in the following, P05 was not sufficient to capture the regional background
concentrations. The color scales in Fig. 5 and 6 represent the averaged enhancement over the
background concentrations of each source/species. For a better visualization, the maximum of
the color scale was set at the 75$^{th}$ percentile (P75) for $SO_4$, $NO_3$, eBC, $CO_2$ and CO.
Moreover, the highest 75$^{th}$ percentile among all OA sources (i.e., 1.2 µg m$^{-3}$ in Tartu and 2.4
µg m$^{-3}$ in Tallinn) was used as a maximum for the four OA sources, in order to facilitate the



comparison among them. Lastly, the sizes of the points represent the number of measurement
points that were averaged in each case. Longitude profiles of the enhancements of all
considered component were obtained for Tartu by averaging the calculated enhancements in
longitude bins (using the same grid of 250 m$^2$ as above). These results are shown in Fig. 7
(averages and standard deviation among all loops), Fig. S8 (median and first and third
quartiles) and Fig. S9 (separation of all loops into time-bins of two hours). The longitude
profiles in Fig. 7 and Fig. S8 allowed for the calculation of regional background
concentrations and urban increments, as defined by Lenschow et al. (2001) and reported in
Table 2. The urban concentrations, which are given by the sum of the regional background
and the urban increment, represent a mix between urban background and kerbside locations.
While the averaged profiles take into account the effects of the measured point sources in the
urban area (mostly traffic and residential emissions), the use of the median profiles is
expected to exclude these effects, making the results more representative of the urban
background concentrations. In the following we will present the results related to the average
profiles, followed by the results from the median profiles reported in parenthesis.  In all
cases, the longitude profiles were fitted using sigmoid functions (black curves). In order to
have a constant averaging city area, the fitting limits (indicated with blue and pink arrows)
and the x-value of the sigmoid's midpoint ($X_0$) were determined from the fit of the total PM$_{2.5}$
mass (NR-PM$_{2.5}$ plus eBC) and imposed to all other components. In most of the cases the
base of the sigmoid functions is slightly above zero. This indicates that the P05 previously
subtracted didn't represent the full regional background, which is therefore given by the sum
of the average P05 and the base of the sigmoid function. Moreover, the fits on the west side
of Tartu show always higher base values than those for the east, indicating the influence of
local sources in the considered regional background area west of Tartu. However, these
differences between the west and east fits are in most cases rather low, and therefore we use
the west-east averages to calculate the urban increments concentrations in Table 2.
In Tartu, the three primary OA sources (HOA, BBOA and RIOA) show a clear enhancement
in the city center compared to the suburban areas (Fig. 7 and S8). Moreover, different source
regions (see Fig. 5a-c) and emission times (see Fig. S9) can be distinguished inside the urban
area. For example, maximum HOA concentrations are observed on highly congested roads,
especially at sites under stop-and-go conditions, and show a maximum enhancement in the
morning and evening traffic rush hours (07:00 to 09:00 and 15:00 to 17:00, LT). The spatial
distributions of the eBC, CO$_2$ and CO (Fig. 5g-i) are consistent with that of HOA, which





indicates that these species originate mostly from traffic. BBOA is more strongly enhanced in
the residential areas and the maximum enhancement is seen in the evening hours (15:00 to
21:00, LT) when domestic heating is more active. RIOA shows enhanced contributions in
both, the residential areas (probably related to domestic cooking emissions) and the major
roads in the city center (probably related to cooking emissions from restaurants). The
maximum enhancement of RIOA is also seen in the evening hours (15:00 to 19:00, LT),
during and after the evening maximum of HOA. In contrast, OOA (Fig. 5d) and the other
secondary species ($SO_4$, $NO_3$, $NH_4$ and Cl, see Fig. 5e-f and Fig. S6), show very
homogeneous spatial distribution over the whole measurement area (as expected from their
secondary nature), and no clear dependence on the time of the day can be seen for the OOA
(Fig. S8). Although slight enhancements are observed in these components close to
residential areas (OOA enhancement of 0.8 µg m$^{-3}$), these increases are negligible within the
measurement and source apportionment uncertainties.
As reported in Table 2, the $PM_{2.5}$ mass concentration in Tartu shows an urban increment of
6.0 (4.6) µg m$^{-3}$ over a regional background concentration of 4.0 (3.5) µg m$^{-3}$. This leads to
urban $PM_{2.5}$ mass concentrations of up to 10 (8.1) µg m$^{-3}$, which represents an increase of a
factor 2.5 (2.3) in the particle mass concentration in the urban area compared to the regional
background. About half of this enhancement is related to the emissions of eBC, which shows
an increase of 3.2 (2.3) µg m$^{-3}$ over a regional background of 1.1 (0.58) µg m$^{-3}$. Thus, the
urban concentration of eBC is 4.2 (2.9) µg m$^{-3}$, which represents an enhancement of a factor
3.9 (5.0) of eBC in the urban area. The primary OA sources explain great part of the
remaining increase in the $PM_{2.5}$ mass: HOA is increased by a factor 3.6 (3.0) in the urban
area and has contribution of 1.7 (1.0) µg m$^{-3}$ to the urban concentration; RIOA is enhanced
by a factor 2.0 (2.3), contributing with 1.7 (1.0) µg m$^{-3}$ to the urban concentration; and
BBOA is enhanced by a factor 3.1 (2.4) and contributes with 1.0 (0.52) µg m$^{-3}$ to the urban
concentrations. On the other hand, OOA and the inorganic species ($SO_4$, $NO_3$, $NH_4$ and Cl)
show very low increases in the urban area, resulting in a total urban increment below 0.21 µg
m$^{-3}$ (average and median). In the gas-phase, $CO_2$ shows an increase of 8.3 (5.3) ppm over a
regional background of 403.5 ppm (both average and median); CO is increased by 0.15 (0.11)
ppm over a regional background of 0.16 (0.14) ppm, which represents an increase of a factor
1.9 (1.7); while $CH_4$ shows very similar concentrations inside and outside the city, with
average (and median) regional background of 1.90 ppm and urban concentrations of 1.91
ppm.



Similar results were obtained for Tallinn (see Fig. 6 and Fig. S7). However, given the larger
extension of this city, it wasn't possible to include a real regional background site in the
route. Therefore, the longitude profiles and urban increments couldn't be properly explored
for Tallinn. However, different source regions can still be distinguished within the examined
area. Thus, the spatial distribution of HOA (Fig. 6a) is in agreement with those of eBC, $CO_2$
and CO (Fig. 6g-i) and shows substantial increases in areas with high traffic and on major
streets in the city center with significant stop-and-go conditions. BBOA (Fig. 6b) has higher
contributions in the two residential areas, while compared to Tartu, in Tallinn the spatial
distribution of RIOA (Fig. 6c) is more homogeneous, with only slight enhancements in the
residential area and in the city center. Finally, OOA (Fig. 6d) exhibits a small enhancement in
the city center area, which again coincides with small increases in the secondary inorganic
species concentrations (see Fig. 6e-f and Fig. S7) that are insignificant within the
measurement and source apportionment uncertainties. Enhanced $SO_4$ levels are also found in
the northern part of the route, likely from local ship emissions (Lack et al., 2009).
**3.4    Special events: transport and accumulation of pollutants**
Enhanced concentrations of secondary species including OOA, $SO_4$, $NO_3$ and $NH_4$ were
measured during the first measurement day in Tartu (see Fig. 2a and Fig. 4a). The analysis of
the 24-hour back-trajectories reported in Fig. 8a indicates that these mostly secondary
components were probably transported from continental Europe, in particular from northern
Germany. The later decrease in the concentrations of these species coincides with clean air
masses originating from the Northern Atlantic at higher altitudes above ground level. As
reported in Fig. 8b, during this transport event the average $PM_{2.5}$ mass concentration
increased to 28.3 µg m$^{-3}$ (compared to average concentrations of 11.0 µg m$^{-3}$ measured
during day-time and 6.5 µg m$^{-3}$ during night-time). This increase in mass is mostly related to
the increased concentrations of the secondary components, especially of $NO_3$ and OOA.
Accordingly, the relative contributions of the inorganic species to the total NR-$PM_{2.5}$
increased to over 44 % during the transport event (compared to 12 % for day-time and around
28 % for night-time averages) and the relative contribution of the OOA to total OA increased
to 56 % (compared to 25 % for day-time and 42 % for night-time averages). It is worth to
note that source separation is more uncertain during the transport event due to lower statistics
and increased mixing (if the transported air contains multiple sources). This is especially the





case for RIOA, which has a relative error of 41 % (estimated by the bootstrapping procedure)
during the transport event.
During the nights of 28 and 29 March, very high concentrations of organics (exceeding 200
µg m$^{-3}$), eBC (above 15 µg m$^{-3}$) and $CO_2$ (up to 500 ppm) were measured in Tallinn, as
shown in Fig. 9a. Relatively short back-trajectories originating from the Baltic Sea (North-
West and West from the sampling site) and at high altitudes were obtained for these periods
(not reported). Moreover, as shown in Fig. 9a, during such accumulation events wind speed
was close to zero and a strong near-ground temperature inversion (i.e. a positive temperature
difference between the ground and 22 m above ground level (AGL)) was observed. Under
such conditions, the vertical mixing is suppressed and the local pollutants are trapped at the
surface. As reported in Fig. 9b, during the accumulation periods the average $PM_{2.5}$ mass
increased up to 41.7 µg m$^{-3}$, with OA explaining 73 % of the total mass. This increase was
mostly related to the increase of the primary aerosols, mainly HOA and BBOA, which
explained 33 and 37 % of the OA mass, respectively.
**4    Conclusions**
Mobile measurements allowed for the study of the spatial distributions of major gas- and
particle-phase pollutants in two urban areas in Estonia, permitting the identification of
particular source areas and the determination of regional background concentrations and
urban increments for the individual components/sources. A comprehensive set of instruments
including a HR-ToF-AMS (with a newly developed inlet to measure the NR-$PM_{2.5}$ fraction),
a 7-wavelength Aethalometer and several gas-phase monitors were deployed in the mobile
laboratory to retrieve a detailed chemical characterization of the $PM_{2.5}$ fraction and the
concentrations of several trace gases with high time resolution.
The measurements were performed in March 2013 in the two major cities of Estonia (Tallinn
and Tartu) and no major differences were found in the chemical composition at the two sites.
Higher mass concentrations were always measured during day-time, when point sources were
sampled during mobile measurements. Under regular meteorological conditions, OA
represented the largest mass fraction (on average 52.2 % to 60.1 % of $PM_{2.5}$), while the
relative contribution of the inorganic species (mostly $SO_4$, $NO_3$ and $NH_4$) strongly increased
during the transport of polluted air masses from northern Germany. Four sources of OA were
identified by means of PMF: three primary sources (HOA, BBOA and RIOA) and a



secondary OA (OOA). Although the RIOA is thought to be dominated by cooking emissions,
contributions from other residential emissions to this factor cannot be excluded. For example,
waste burning is known to be a common process in some cities in Estonia (Maasikmets et al.,
2015). However, to properly separate the contribution of waste burning from other co-
emitting sources, laboratory studies of direct emissions need to be performed in the future.
While OOA dominated the OA mass during night-time (on average 42.3 % in Tartu and 43.8
% in Tallinn), the primary sources explained the major fraction of OA during day-time (75.2
% in Tartu and 68.3 % in Tallinn, with similar contributions from the three sources). During
the period with transport of polluted air masses aforementioned, the OOA relative
contribution was enhanced. In contrast, HOA, RIOA and BBOA were strongly enhanced
during periods characterized by temperature inversions, which induced the accumulation of
locally emitted primary pollutants (primary OA and eBC).
Different source regions were identified inside the two urban areas. All traffic related
pollutants (including HOA, eBC, $CO_2$ and CO) where strongly enhanced on the major city
roads, especially in areas with stop-and-go conditions during the morning and evening rush
hours. BBOA showed a clear increase in the residential areas during the evening hours (due
to domestic heating), while RIOA (believed to be strongly influenced by cooking emissions)
was enhanced in both, the city center (from restaurant cooking emissions) and in the
residential areas (from domestic cooking). In contrast, the secondary components (including
OOA, $SO_4$, $NO_3$, $NH_4$ and Cl) had very homogeneous spatial distributions, with no clear
enhancement in the urban areas (within the measurement uncertainties) or at certain times of
the day. For Tartu, regional background concentrations and urban increments of all measured
components/sources were also determined. On average, the $PM_{2.5}$ mass had an enhancement
inside the city of 6.0 µg m$^{-3}$ over the regional background concentration of 4.0 µg m$^{-3}$. This
urban increment was strongly related to the enhancement of eBC (3.2 µg m$^{-3}$) and the
primary OA sources (on average 1.2 µg m$^{-3}$ from HOA, 0.67 µg m$^{-3}$ from BBOA and 0.72 µg
m$^{-3}$ from RIOA), while the secondary components (OOA, $SO_4$, $NO_3$, $NH_4$ and Cl) didn't
contribute to a substantial enhancement. Moreover, the good correlation found between eBC
with HOA indicates that up to 74 % of the enhancement in the $PM_{2.5}$ is related to traffic
emissions in the urban area. $CO_2$ and CO, which were also found to be strongly correlated
with HOA, had an average urban increment of 8.3 and 0.15 ppm over regional background
concentrations of 403.5 and 0.15 ppm, respectively.





Our results show that mobile measurements are a very powerful technique for spatial
characterization of the major pollutants in urban areas. The methodology presented in this
work can be generalized to other cities, in order to determine the influence of human activity
on the particle sources and levels in different areas of a city and the related health effects.
**Acknowledgments**
This work was carried out in the framework of the public procurement "Determination of
Chemical Composition of Atmospheric Gases and Aerosols in Estonia" of the Estonian
Environmental Research Centre (Reference number: 146623), funded by the Estonian-Swiss
cooperation program "Enforcement of the surveillance network of the Estonian air quality:
Determination of origin of fine particles in Estonia". JGS acknowledges the support of the
Swiss National Science Foundation (Starting Grant No. BSSGI0 155846). IEH acknowledges
the support of the Swiss National Science Foundation (IZERZ0 142146). The authors
gratefully acknowledge the NOAA Air Resources Laboratory (ARL) for the provision of the
HYSPLIT transport and dispersion model and READY website (http://www.ready.noaa.gov)
used in this publication.



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

exponents for traffic and wood burning in the Aethalometer based source apportionment
using radiocarbon measurements of ambient aerosol, In preparation.




1   Table 1: Instrument list, measured components and time resolution of each measurement.

| | Instrument list | Measured components | Time resolution |
|---|---|---|---|
| **Aerosols** | **HR-ToF-AMS** | *Size resolved chemical composition of NR-PM2.5* | *25 sec* |
| | **Aethalometer** | *BC (7λ)* | *1 sec* |
| **Gas** | **CO₂ Picarro** | *CO₂, CO, CH₄, H₂O* | *1 sec* |
| | **CO₂ Licor** | *CO₂, H₂O* | *1 sec* |
| **Others** | **GPS, Temperature, Relative humidity & Solar radiation** | | *2 sec* |





Table 2: Results obtained from the average (A) and median (B) longitude profiles for each
measured component/source. P05 represents the averaged 5[th] percentile subtracted for the
calculation of the enhancements; base and increment values were obtained from the sigmoid
fits; the regional background is given as the sum of P05 and the average base value; urban
concentrations are the sum of the regional background and the average urban increment; the
factor increase represents the ratio between the urban and the regional backgrounds.
**(A) Average longitude profiles:**

| | P05[(1)] | Base | | | Urban increment | | | Regional background | Urban concentration | Factor increase |
|---|---|---|---|---|---|---|---|---|---|---|
| | | West | East | Average | West | East | Average | | | |
| $PM_{2.5}$ (µg m$^{-3}$) | 1.8 | 2.6 | 1.8 | 2.2 | 5.6 | 6.3 | 6.0 | **4.0** | **10.0** | **2.5** |
| HOA (µg m-3) | 0.18 | 0.34 | 0.24 | 0.29 | 1.2 | 1.3 | 1.2 | **0.47** | **1.7** | **3.6** |
| BBOA[(2)] (µg m-3) | 0.11 | 0.24 (0.16) | 0.19 | 0.21 | 0.60 (0.64) | 0.75 | 0.67 | **0.32** | **1.0** | **3.1** |
| RIOA (µg m-3) | 0.27 | 0.44 | -0.30 | 0.44 | 0.72 | 1.9 | 0.72 | **0.71** | **1.4** | **2.0** |
| OOA (µg m-3) | 0.44 | 0.42 | 0.32 | 0.37 | 0.024 | 0.11 | 0.069 | **0.81** | **0.87** | **1.1** |
| $SO_4$ (µg m-3) | 0.29 | 0.075 | 0.055 | 0.065 | 0.032 | 0.051 | 0.042 | **0.35** | **0.39** | **1.1** |
| $NO_3$ (µg m-3) | 0.095 | 0.075 | 0.076 | 0.075 | 0.042 | 0.038 | 0.040 | **0.17** | **0.21** | **1.2** |
| $NH_4$ (µg m-3) | 0.079 | 0.032 | 0.028 | 0.030 | 0.012 | 0.016 | 0.014 | **0.11** | **0.12** | **1.1** |
| Cl (µg m-3) | 0.012 | 0.036 | 0.035 | 0.035 | 0.022 | 0.022 | 0.022 | **0.047** | **0.069** | **1.5** |
| eBC (µg m-3) | 0.34 | 0.96 | 0.54 | 0.75 | 3.0 | 3.3 | 3.2 | **1.1** | **4.2** | **3.9** |
| $CO_2$ (ppm) | 403.0 | 0.99 | 0.04 | 0.52 | 7.8 | 8.9 | 8.3 | **403.5** | **411.9** | **1.0** |
| CO (ppm) | 0.14 | 0.028 | 0.012 | 0.020 | 0.14 | 0.15 | 0.15 | **0.16** | **0.31** | **1.9** |
| $CH_4$[(3)] (ppm) | 1.90 | 0.0060 (0.0052) | <0.001 | 0.0012 | 0.0047 (0.0064) | 0.012 | 0.0083 | **1.90** | **1.91** | **1.0** |

**(B) Median longitude profiles:**

| | P05[(1)] | Base | | | Urban increment | | | Regional background | Urban concentration | Factor increase |
|---|---|---|---|---|---|---|---|---|---|---|
| | | West | East | Average | West | East | Average | | | |
| $PM_{2.5}$ (µg m$^{-3}$) | 1.8 | 1.8 | 1.6 | 1.7 | 4.6 | 4.6 | 4.6 | **3.5** | **8.1** | **2.3** |
| HOA (µg m-3) | 0.18 | 0.16 | 0.13 | 0.14 | 0.66 | 0.66 | 0.66 | **0.33** | **1.0** | **3.0** |
| BBOA (µg m$^{-3}$) | 0.11 | 0.088 | 0.12 | 0.11 | 0.35 | 0.27 | 0.31 | **0.22** | **0.52** | **2.4** |
| RIOA (µg m-3) | 0.27 | 0.20 | 0.15 | 0.17 | 0.58 | 0.60 | 0.59 | **0.45** | **1.0** | **2.3** |
| OOA (µg m-3) | 0.44 | 0.28 | 0.26 | 0.27 | 0.084 | 0.096 | 0.090 | **0.71** | **0.80** | **1.1** |
| $SO_4$ (µg m-3) | 0.29 | 0.064 | 0.053 | 0.059 | 0.029 | 0.039 | 0.034 | **0.35** | **0.38** | **1.1** |
| $NO_3$ (µg m-3) | 0.095 | 0.043 | 0.053 | 0.048 | 0.056 | 0.039 | 0.047 | **0.14** | **0.19** | **1.3** |
| $NH_4$ (µg m-3) | 0.079 | 0.028 | 0.026 | 0.027 | 0.0094 | 0.011 | 0.010 | **0.11** | **0.12** | **1.1** |
| Cl (µg m-3) | 0.012 | 0.022 | 0.025 | 0.024 | 0.024 | 0.019 | 0.021 | **0.035** | **0.06** | **1.6** |
| eBC (µg m-3) | 0.34 | 0.45 | 0.027 | 0.24 | 2.0 | 2.5 | 2.3 | **0.58** | **2.9** | **5.0** |
| $CO_2$ (ppm) | 403.0 | 0.95 | 0.051 | 0.50 | 5.0 | 5.6 | 5.3 | **403.5** | **408.8** | **1.0** |
| CO (ppm) | 0.14 | 0.011 | <0.001 | 0.0052 | 0.096 | 0.12 | 0.11 | **0.14** | **0.25** | **1.7** |
| $CH_4$[(3)] (ppm) | 1.90 | 0.0032 (0.0028) | <0.001 | <0.001 | 0.0051 (0.0055) | 0.011 | 0.0079 | **1.90** | **1.91** | **1.0** |

*(1) Excluding special events   (2) ($X_0$ not fixed)   (3) Excluding spike*





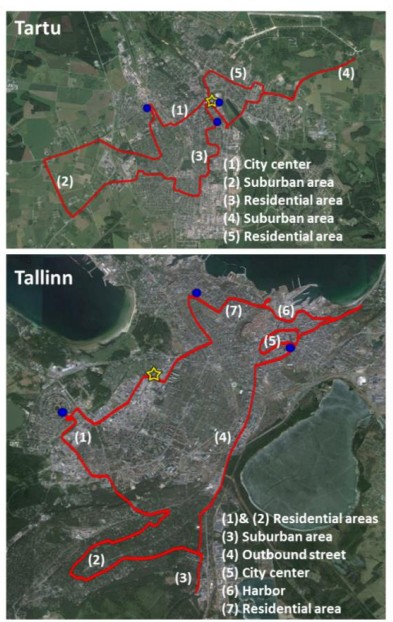

2  Figure 1: Driving routes in Tartu (top) and Tallinn (bottom). Red line represents: GPS data;

3  Yellow star: stationary measurements location; Blue dots: monitoring stations of the Estonian

4  Environmental Research Institute (EERC).



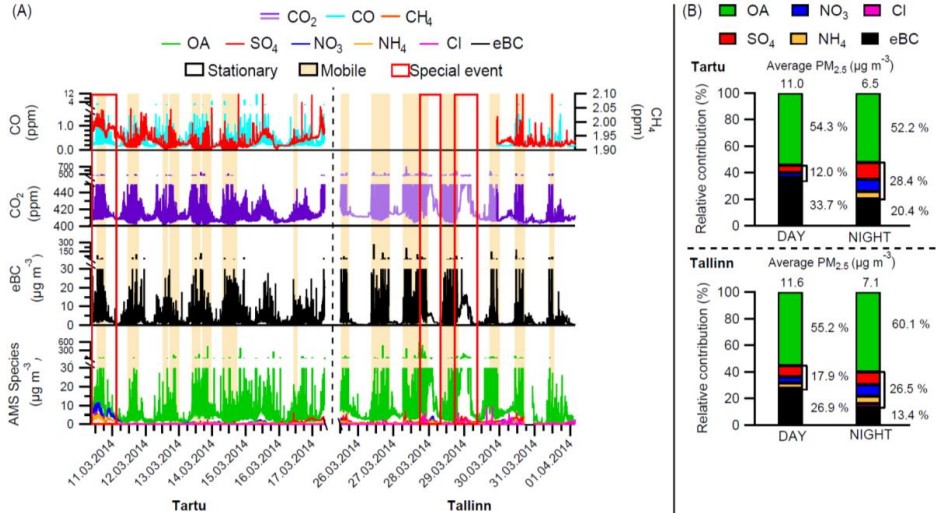

Figure 2: (a) Temporal evolution of all gas- and particle-phase measured components over
the full measurement period; (b) Average $PM_{2.5}$ (NR-$PM_{2.5}$ plus eBC) mass concentration and
chemical composition for the measurements in Tartu (top panel) and Tallinn (bottom panel),
with day- and night-time distinction.





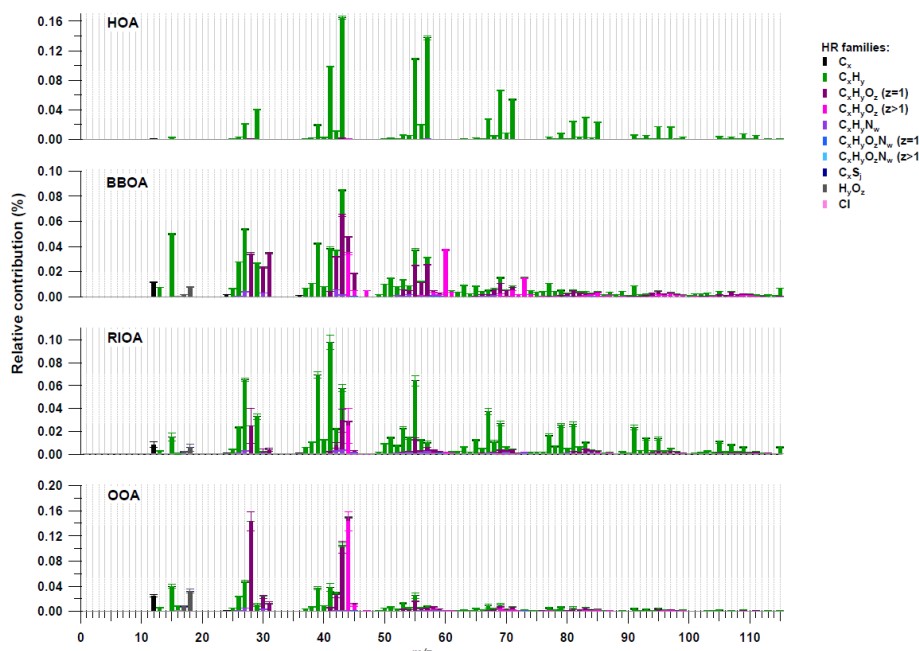

2 Figure 3: Mass spectra of the four OA sources identified with PMF. From top to bottom:

3 HOA, BBOA, RIOA and OOA. Error bars indicate the standard deviation among 100

4 bootstrap runs.





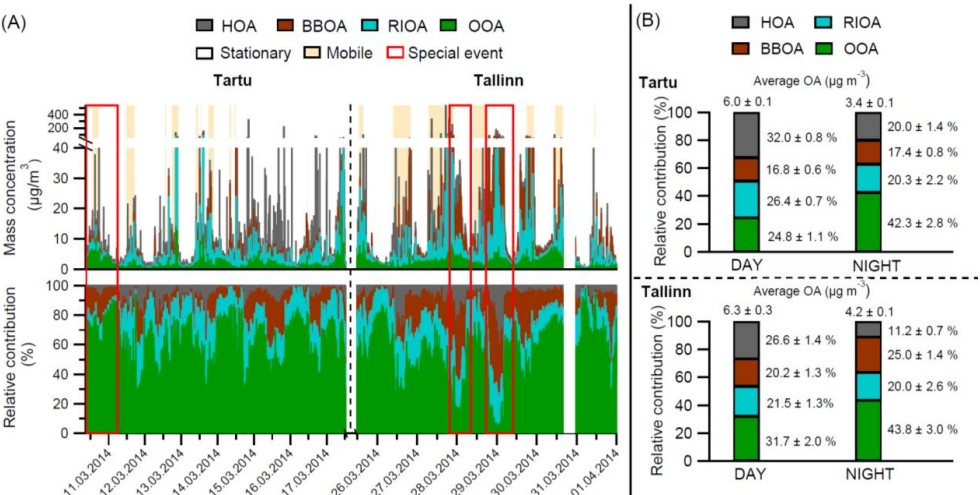

Figure 4: (a) Temporal evolution of the absolute mass (top panel) and relative contributions
(bottom panel) of the four OA sources over the full measurement period; (b) Average OA
mass concentrations and relative contributions of the OA sources for the measurements in
Tartu (top panel) and Tallinn (bottom panel), with day- and night-time distinction. Errors
indicate the standard deviation among 100 bootstrap runs.



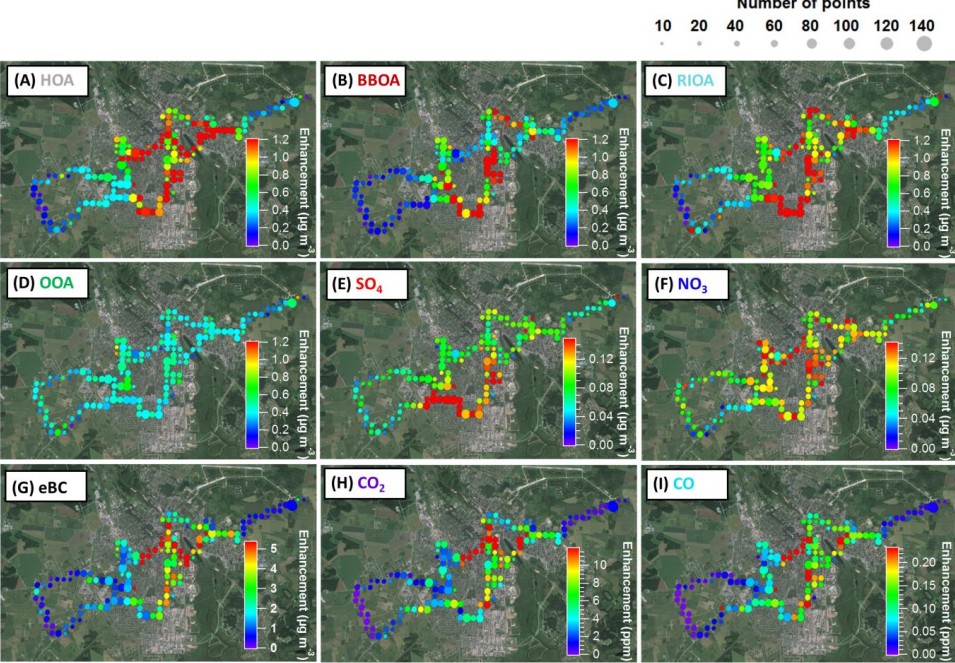

Figure 5: Average spatial distributions of all identified OA sources (panels a-d) and other
measured components (panels e-i) in Tartu. The color scales represent enhancement over the
background concentrations; the maximum of the color scales have been fixed to the 75[th]
percentile of the average enhancement of each component in panels e-i and to the highest 75[th]
percentile among all OA sources in panels a-d. The sizes of the points represent the number
of points that have been averaged in each case.

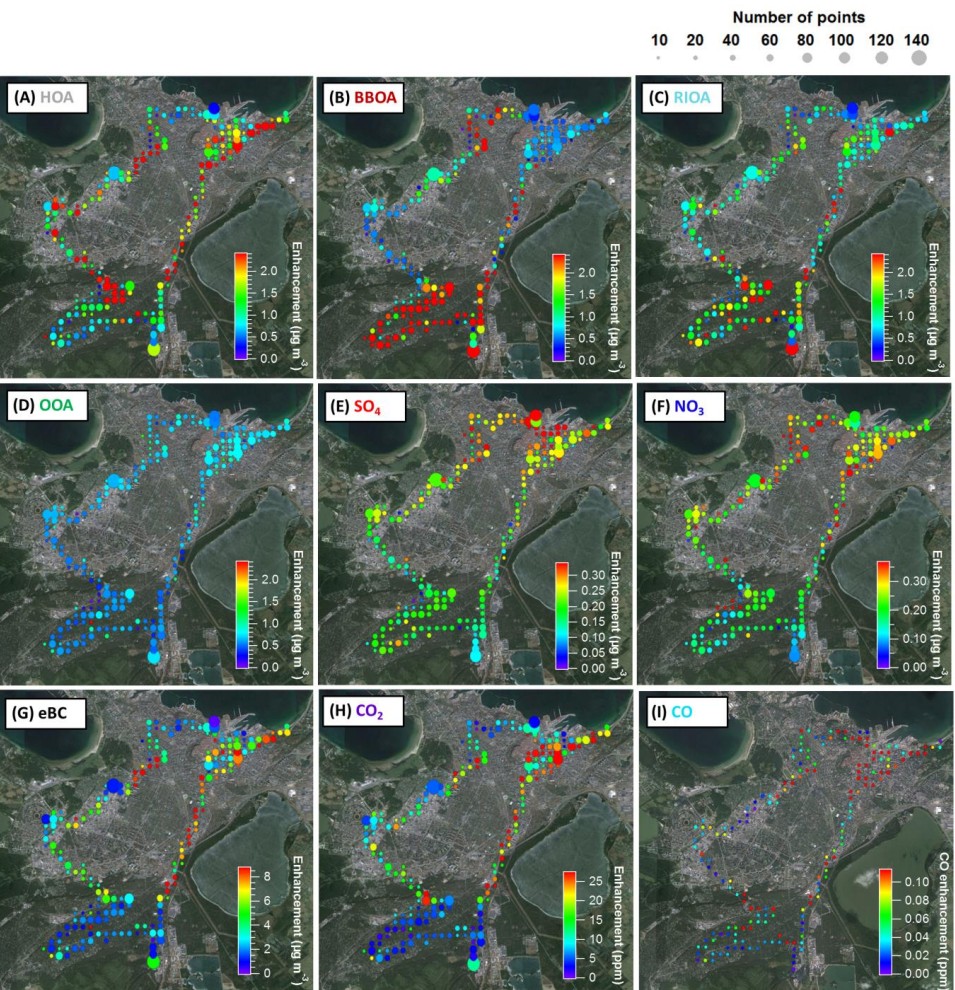

Figure 6: Average spatial distributions of all identified OA sources (panels a-d) and other
measured components (panels e-i) in Tallinn. The color scales represent enhancement over
the background concentrations; the maximum of the color scales have been fixed to the 75[th]
percentile of the average enhancement of each component in panels e-i and to the highest 75[th]
percentile among all OA sources in panels a-d. The sizes of the points represent the number
of points that have been averaged in each case (Note: less data available for CO).

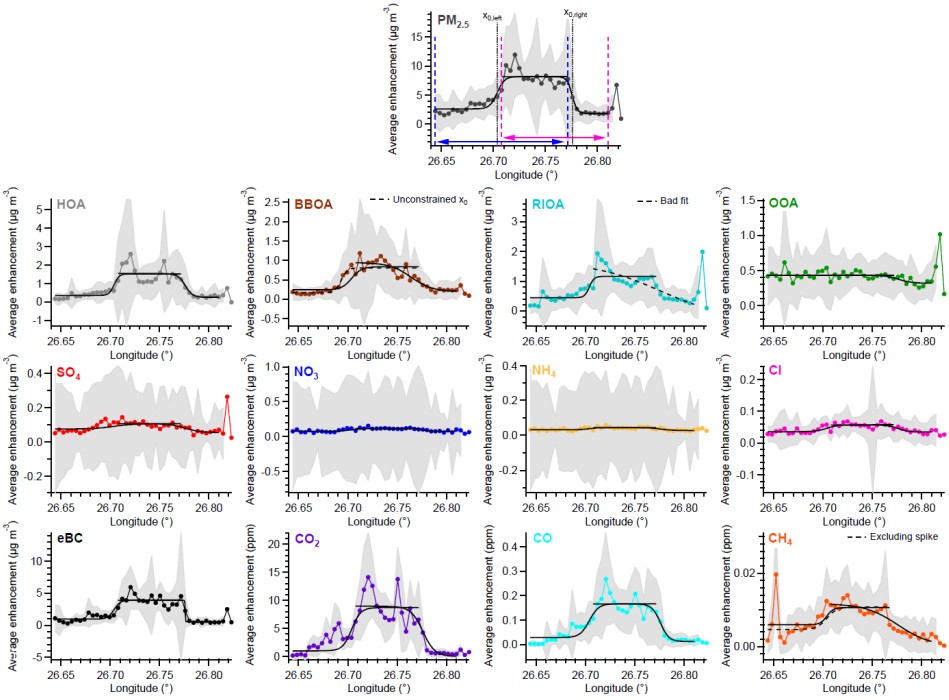

Figure 7: Average longitude profiles of the enhancements of all measured components and
sources in Tartu. Colored curves represent the average enhancement of each
source/components over 26 loops and the grey shaded area is the standard deviation among
them. The average enhancements were fitted with sigmoid functions (black curves). The
fitting limits (pink and blue arrows in top panel) and the sigmoid's midpoint ($X_0$) were
determined from the fit of the total $PM_{2.5}$ mass (NR-$PM_{2.5}$ plus eBC) and then imposed to the
other components/sources. Dashed black lines indicate a non-standard fit (described in each
case in the plot) and the results of these fits are represented in parenthesis and grey color in
Table 2. Notes: The spike found in the east for RIOA, OOA and $SO_4$ is not representative, as
it is related to one single measurement point. The spike in $CH_4$ in the west side is related to
consistent increases of this component nearby a cowshed and will be further investigated in a
future publication.





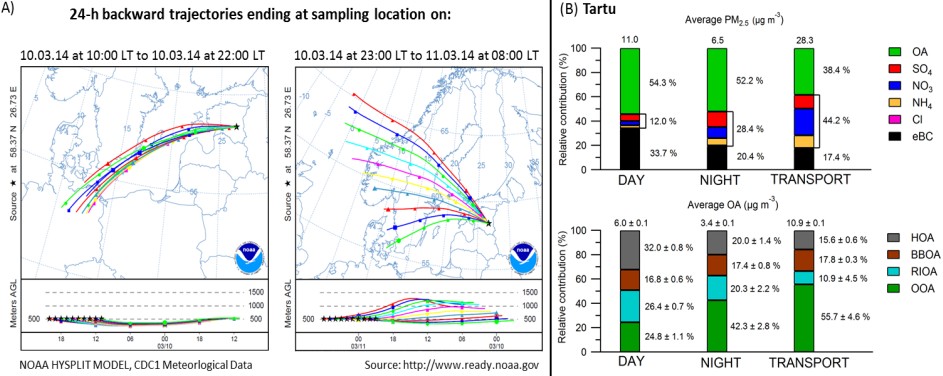

Figure 8: (a) 24-hour back-trajectories (NOAA HYSPLIT MODEL) of the air masses ending
at the sampling location (Tartu) during the transport event (left panel) and the successive
hours (right panel). (b) PM$_{2.5}$ mass concentration and chemical composition (top panel) and
OA mass concentration and relative contributions of the OA sources (bottom panel) during
the measurements in Tartu during day-time, night-time and transport event. Errors indicate
the standard deviation among 100 bootstrap runs.



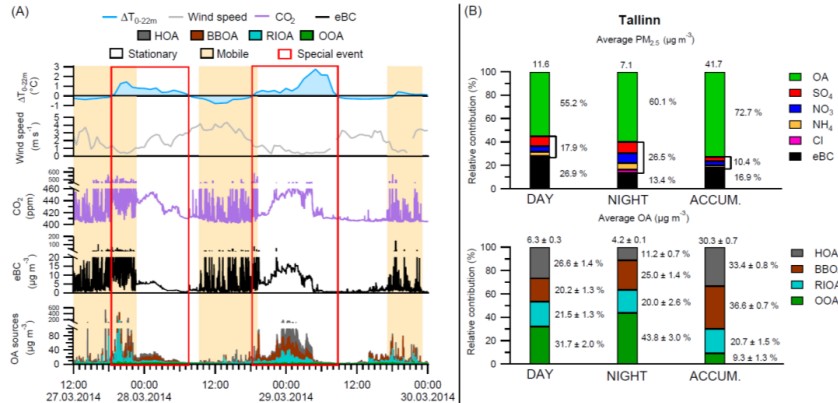

Figure 9: (a) Temporal evolution of the OA sources, eBC and $CO_2$, wind speed and $\Delta T_{0\text{-}22m}$
(temperature difference between ground level and at 22 meters above ground level) during
the accumulation events in Tallinn. (b) $PM_{2.5}$ mass concentration and chemical composition
(top panel) and OA mass concentration and relative contributions of the OA sources (bottom
panel) during the measurements in Tallinn during day-time, night-time and accumulation
events. Errors indicate the standard deviation among 100 bootstrap runs.