# Peer review of "Urban increments of gaseous and aerosol pollutants and their sources using mobile aerosol mass spectrometry measurements"

_Atmospheric Chemistry and Physics, 2016_

## Referee Comment (RC1) · Anonymous Referee #3 · 23 Feb 2016

The paper by Else et al. summarizes mobile springtime measurements of aerosol concentrations and several gas phase species in two Estonian cities. The measurements allowed the authors to identify 4 classes of OA in both cities. Overall, aerosol composition in both cities was similar and was dominated by higher concentration of primary types of OA during the day and by lower amount of secondary OA at night. Contribution of the secondary inorganic species was low except during a transport event.

The manuscript is very well-organized and well-written. There are two aspects that need some work in my opinion. One is related to wind direction and its variability during day and night sampling and how it might affect the interpretation of the results (see my comment below). The other aspect is that since the measurement was done

in two cities, I think more can be done to compare quantitatively aerosol air quality in these two cities. Since measurements of CO are already available, I think it will be valuable to look at the enhancement ratios (not by subtracting a background) but considering scattering plots of say OA vs. CO, BC vs CO (or the PMF-resolved factors or other species vs. CO) in comparable times of the day to separate out the differences in dynamics, boundary layer heights, dilution, etc. and be able to determine a more valuable comparison of the aerosol sources in these cities. This will also allow the authors to compare the measurements with other measurements (ground based on airborne) in other cities around the world. I support publishing the paper after my comments (above and below) are addressed.

Abstract: indicate which month/season the measurements were carried out.

P5, L3: It is mentioned that stationary measurements were made at night. Were there any mobile measurements also carried out at night?

P8, L2: Explain why A(abs)=1.7 was used for wood burning BC? And why was it that the lower wavelength of 370 nm was not used? Doesn't it make sense to use 370 nm since BrC would be stronger there?

P8, L16-17: Just looking at Figure 2, it seems standard deviations of the averages would be really high, and maybe that's why they're not indicated along with the average values in Panel B. I wonder if estimates of the median values (or to be more complete, box and whisker plots) of the tracers will be more valuable than the average values.

P11, L15: missing a word ". . .of ?? (data??)..."

P12, L1: replace kurbside with curbside

Figure 2: I suggest having the inorganics on a separate axis, with max ∼10 ug/m3, so you can see the tracers better.

Figure 5: For some species, it appears that the conc. were very different on different sides of the loop, suggesting that the sources are towards the center of the loop (as

opposed to one side, e.g., BBOA and sulfate). In other words there is gradient in the latitudinal direction as well as longitudinal direction. To further investigate the source regions, it makes sense to consider wind direction data with these distribution maps. Were wind directions consistent during the day and night sampling time? It seems the averages include both daytime and nighttime. Could you add average wind barbs representatives for daytime and nighttime or at least discuss the wind patterns in the text? Correct interpretation of the mean and median values in Table 2 with relation to the source regions also needs some knowledge of the wind direction.

Figure 7: Indicate in the legend that enhancement is relative to P05 values.

Table 2 legend: Indicate which city the stats refer to.

---

## Referee Comment (RC2) · Anonymous Referee #2 · 26 Feb 2016

**General Comments:**

Elser et al. describe gas and aerosol measurements conducted in two Estonian cities. The authors use a mobile platform to investigate the extent to which pollution concentrations within the city limits exceed regional background levels. Via source apportionment, the authors attribute the observed organic aerosol loadings to four primary sources: traffic emissions, biomass burning, primary residential emissions, and secondary aerosol formation. The authors map the spatial distribution of each component to identify source "hot spots." In both cities, traffic-related components were most variable in the city center while biomass burning and primary residential emissions were concentrated in populated regions. Secondary components were well distributed throughout each city and had the least impact by point sources; however, increases in secondary aerosol were most strongly influenced by long-range transport of air masses originating from polluted regions west of Estonia.

The paper presents a useful and effective methodology for studying the impact of point sources on local air quality. Furthermore, these measurements are important as they assess the pollutant levels and source contributions in an understudied region of Europe. The manuscript is very well organized and many of the conclusions are well supported. I have some recommendations that would improve the quality of the manuscript. Upon addressing these comments, I recommend the manuscript for publication.

First, I believe that further details of the source apportionment should be included to strengthen the argument of a four-factor solution. Questions and comments pertaining to this aspect of the manuscript are summarized below. Second, parts of the methodology section employ sentence structure and wording that, at times, is awkward and/or difficult to follow. While I do not wish to interfere with stylistic choices, I believe that some rewording of these sections may help the manuscript read more fluently. Suggestions are provided in the Minor Comments.

*Source Apportionment*

The authors identify four factors that sufficiently describe the variation in the data. The authors are thorough with the comparison of factors with external tracers; however, there is little discussion and no figures demonstrating the model residuals as the PMF solution is pushed to higher factors. The authors describe their observations (Page 10, lines 12 – 24), however a figure should be included demonstrating the behavior of $Q/Q_{exp}$ as a function of the number of factors. Furthermore, the authors present a 5-factor solution, but argue that the fifth "unknown" factor exhibits a primary emission temporal pattern (which is uncharacteristic of a LV-OOA factor) and therefore does not significantly improve the interpretation of data. While this may be true, I believe it is necessary to demonstrate that the temporal residuals are not significantly improved for a 5-factor solution. It may be that the "unknown" factor results from factor splitting or some other mathematical construct. In any case, the PMF discussion should better describe the factor residuals.

The authors invoke bootstrapping as a means of constraining the error in the PMF solution. The author's note that bootstrapping inherently varies the algorithm starting point (i.e., seed) and therefore accounts for model uncertainties; however, the PMF solution may also be strongly affected by variations in fpeak (Ulbrich et al. 2009). There is little discussion about the rotational ambiguity of the PMF solution. I believe this discussion is necessary in order to evaluate the robustness of the PMF solution.

Finally, I believe it would be useful to compare the factor profiles to published spectra. This comparison would provide additional justification for the resolved factors. Specifically, I have some questions regarding the RIOA factor. The RIOA factor exhibits a temporal pattern that appears to be unique; however, the RIOA factor only exists in a 4 or higher factor solution and is primarily resolved from the BBOA and HOA factors (Fig S3). Consequently, the factor associated with RIOA results from the contribution of two other primary emission factors. While this may be simply due to the fact that

RIOA, BBOA, and HOA are common in residential areas, this result may also be a result of factor splitting.

There are a number of ways the authors can provide additional evidence in support of the RIOA factor. The simplest option would be to compare the factor profiles to published spectra. The authors provide some comparison in the text, however a supplemental figure would be more illustrative. A more thorough analysis would be to perform PMF on subsets of the data and determine if the RIOA factor is still resolved. For example, if one were to remove time periods when the RIOA component is dominant, does PMF still resolve an RIOA factor? I believe these additional tests would strengthen the authors' PMF solution.

**Specific Comments**:

Page 4, Lines 29-30: Are there any sources that outline the spatial distribution of heating systems within the city? For example, can the authors comment on why BBOA emissions are higher and more dispersed in region (2) of Tallin (Fig 1) as opposed to region (7)?

Page 8, Line 5: What studies have used the eBC source apportionment method? Please provide references.

Page 12, Lines 24-26: From what directions do emissions in Tartu/Tallin drain? It would seem to make most sense to take the upwind concentrations as your regional background. Perhaps a discussion of the topography and typical springtime meteorological conditions would help orient the reader to understand which airspaces reflect background conditions.

Page13, Lines 1-3: Here, you state that BBOA is most enhanced during the evening hours, while on Page 11, Line 7 you state that higher loadings during the day are attributed to an increase primary sources (including BBOA). These statements tend to contradict each other. Please clarify.

Page 13, Lines 1-13: Here, you discuss diurnal patterns. If possible (perhaps with the stationary measurements), it would be most illustrative if the diurnal patterns were included as a figure.

Figures 5 and 6: Consider adding the labeling from Fig 1 to these plots in order to facilitate the identification of source regions.

**Minor Comments**

The following are wording suggestions that may help improve the fluency of the methodology section

*Section 2.1*

Page 4, Line 13: "… Tartu, with an area of 38.8 km$^2$…"

Page 4, Line 21: "…to strongly enhance the signal of traffic emissions …"

Page 4, Line 23: "… with low stacks , a detailed…."

*Section 2.2*

Page 5, Line 17: "…For this work , the AMS …"

Page 5, Line 20: "…lens efficiently transmits particles with 80 nm < D$_p$ ≤ 3 µm and has been tested in previous chamber and ambient studies (Williams et al., 2013; Wolf et al., 2015; Elser et al., 2015)"

Page 6, Line 1: "… measurement method automatically corrects for the loading effect …"
The following are additional comments related to the manuscript.

Page 6, Lines 3-6: " The concentrations of trace gases were measured by a Picarro-G2301 $CO_2$/$CH_4$ analyzer and a Licor-6262 CO monitor. "

*Section 2.3*

Page 6, Lines 15-16: "…collection efficiency (CE) algorithm  was used in the calculation of ambient mass concentrations (Middlebrook et al., 2012)."

*Section 2.4*

Page 4, Line 24: "…allows the representation of a two-dimensional …"

Page 7, line 4: "In our case, the model input are the data and error matrices of OA…"

Page7, Line 6: "…contain the fits to the high-resolution data (292 ions)…"

Page 7, Line 7: "…agrees with the mass calculated from the unit mass resolution integration…"

Page 7, Line 13: "…directly calculated from the $CO_2^+$ fragment using the organic …"

Page 7, Line 14: "… were excluded from the PMF analysis…"

Page 7, Line 15: "… variability of the $CO_2^+$; these ions were reinserted post-analysis"

Page 7, Line 18: "… replicate datasets resulting from the perturbation of the original data…"

Page 7, Line 20: " … while other rows are removed (Paatero et al., 2014)…"

Page 7, Line 23: "Note that  each bootstrap…"

Page 7, Line 24: "… initialization point; thus, this methodology inherently includes the investigation of the classic seed variability…"

Page 7, Line 25: "…consistent, suggesting that the solution is robust."

*Section 2.4.2*

Page 8, Line 8: "… for the correlations with the external tracers, …"

---

## Referee Comment (RC3) · Anonymous Referee #4 · 29 Feb 2016

Review of
**"Urban increments of gaseous and aerosol pollutants and their sources using mobile aerosol mass spectrometry measurements"**
By Elser et al.

General comments
This paper presents mobile measurements of gas- and particle-phase pollutants at two Estonian cities of Tartu and Talinn. Detail chemical composition of NR-PM$_{2.5}$ as well as BC and trace gases (CO, CO$_2$, and CH4) were observed in high time resolution and OA characteristics were found to be similar (HOA, BBOA, RIOA, and OOA) at both cities. Primary types of OA (HOA, BBOA and RIOA) were high during the day, whereas secondary OA (OOA) enhanced at night. In summary, the mobile measurements allowed the authors to observe time and spatial distribution of primary and secondary pollutants, as well as influences of long-range transport and local temperature inversion to aerosol and gases pollutants in Tartu and Talinn.

The manuscript is well structured and written. I have some comments regarding research methods, results and discussion as well as suggestions that would improve quality of the manuscript. There are also some typing errors in the text, tables and figures summarized in technical comments. Overall, I support publication of the manuscript after my comments are addressed.

Specific comments
*OA source apportionment:*
I believe the authors have done rigorous analysis in selecting the best PMF factor solution. I expect some of the analysis figures and/or tables are provided in the SI. PMF diagnostic plots, such as those presented by Zhang et al. (2011), are important for understanding the analysis and discussion. Residuals of time series and mass spectra for 3-, 4-, and 5-factor solutions and correlations between factors time series and external tracers (i.e., CO, eBC, SO$_4$, NO$_3$) are useful in understanding selection of the best factor solution. I recommend adding this information in the SI at the least.

*eBC source apportionment:*
It is not clear why do the authors choose to calculate Angstrom exponent using absorption at 470 and 950 nm. Since Zotter et al. paper has not yet published, I could not verify how the calculation was done. I suggest adding a brief description of this calculation in the main text or SI. Also, brief description of calculation of eBC$_{wb}$ and eBC$_{tr}$ would be a useful addition in the SI.

*Results and discussion:*
Ln 1-2: Could the authors provide approximate uncertainty value of ratio of BBOA and eBC$_{wb}$ relative to change in Angstrom exponent?
Ln 2-10: From information provided, it seems reasonable to assign RIOA as COA. The lack of diurnal variability does not mean that the factor is not a COA factor. It is possible that the lack of diurnal variability is due to homogeneous source of cooking emission, or stagnant atmosphere during measurements Have the authors look at meteorological conditions, such as wind direction and speed, when RIOA concentrations were high? I would recommend adding meteorological information in the SI.

Identification of RIOA or COA could be further assessed by plotting the factor mass spectra side-by-side with reference mass spectra from previous studies, such as by Mohr et al. (2012). The authors will need to provide more evidence to support identification of the RIOA factor.

Ln 16-24: The 5[th] factor, LV-OOA, may not show certain process or source, but it can show whether highly oxidized OA is important in the study locations. The LV-OOA can be formed from oxidation of primary OA (e.g., BBOA) or transported into the study location. The authors can discuss this further.

Page 11 Ln 12-15: Increase of RIOA in Talinn is relatively small. I think distribution of RIOA in Talinn is more homogeneous compared to Tartu. Thus, enhancement of RIOA in urban area of Talinn is not well supported.

Page 13 Ln 1-3: Spatial distribution of eBC, CO, and $CO_2$ are consistent not only with HOA but also with BBOA. Thus, they may come from BBOA as well. It would be easier to show consistency or inconsistency by correlation coefficient ($R^2$) between those tracers and HOA and BBOA. Also, in general I disagree that $CO_2$ is mostly traffic because it can be emitted from vegetation and other sources. The authors will need to provide more evidence to support $CO_2$ from traffic.

Technical comments
Page 4 Ln 22: Add reference for these statements.

Ln 10: Unit for flow rate is $m^3 s^{-1}$ or $L min^{-1}$
Ln 12: What is the size of particle in the aerosol inlet before particles are divided into different aerosol measurements.
Ln 28: eBC has been defined in the introduction.

Page 8 Ln 28-30: The statement about enhancement of negative health impacts is not well supported, as it was not within the scope of this study. I suggest the authors to omit the part or revise the sentence.

Page 10 Ln 29: Delete "secondary" or change it to "secondary source (OOA)"

Page 11 Ln 24: "the 5[th] percentile (P05) "

Page 13 Ln 20: Add space "… is 4.2 …"

Table 2: Superscript for the unit $\mu g m^{-3}$.

Figure 2:
(a) What does the different shade of purple for $CO_2$ mean?
(b) If the average pollutant concentrations exclude those from special events, this needs to be included in the title.

Figure 3: I think mass spectra relative contribution is not in %. For comparison, relative contribution in Figure S3 is unitless.

Figure 8: Add in the title that the back-trajectories is from 10:00 at 10 March 2014 to 8:00 at 11 March 2014.

Reference

Mohr, C., DeCarlo, P.F., Heringa, M.F., Chirico, R., Slowik, J.G., Richter, R., Reche, C., Alastuey, A., Querol, X., Seco, R., Penuelas, J., Jimenez, J.L., Crippa, M., Zimmermann, R., Baltensperger, U. and Prevot, A.S.H.: Identification and quantification of organic aerosol from cooking and other sources in Barcelona using aerosol mass spectrometer data, Atmos. Chem. Phys., 12, 1649-1665, doi:10.5194/acp-12-1649-2012, 2012.

Zhang, Q. Q.: Understanding atmospheric organic aerosols via factor analysis of aerosol mass spectrometry: a review, Analyt. Bioanalyt. Chem., 401, 3045-3067, 2011.

---

## Referee Comment (RC4) · Anonymous Referee #1 · 29 Feb 2016

This paper uses mobile measurements to spatially map aerosols and trace gases in two Estonian cities. The use of high time resolution instruments means that good spatial resolution is obtainable and the results are systematically analysed to present statistics on the urban increments. The use of source apportionment techniques on the AMS and Aethalometer data adds extra depth to the results. Overall, the analysis appears solid and the results are well presented. I recommend publication subject to minor (mostly technical) revisions.

Page 1, line 1: Please be more specific when referring to 'polluted continental areas'.

Page 2, line 18: PM2.5 has been the major focus for over 20 years now, so can hardly be described as 'recent'.

Page 6, line 20: Remove the word 'highly'.

Page 11: More explanation and justification of the P05 method should be given. Why was the 5th percentile chosen? What specific effect was the subtraction expected to achieve? Given that the base of the sigmoid fits was allowed to vary, why is this even necessary?

Page 12, line 13: The use of the median does not completely rule out the influence of the kerbside increment because street canyon effects (e.g. when the local wind is perpendicular to a road) can cause on-road emissions to persist is this microenvironment and the increment to no longer manifest itself as discrete spikes in the data. This would cause the median to increase over what would be expected of the urban background. Because these measurements are taken on-road, it is perhaps inevitable that estimates of the urban background will be biased slightly high because of the influence of traffic sources, so this should be added as a caveat. It may be possible to exclude this by selectively averaging the less-busy roads.

Page 12, line 3: Replace 'component' with 'components'.

Page 15: How was the temperature difference between 0 and 22m measured?

––––––––––––––––––––––––

---

## Author Comment (AC1) · 18 May 2016

**Author's response:**

We thank the reviewers for a careful reading and correction of our manuscript. Their suggestions have strongly helped improving the quality of the manuscript.

Following the suggestions of the anonymous referees 2, 3 and 4 we have added in the revised manuscript the description of the meteorological conditions during the measurement periods in both cities. A figure with the time series of wind direction and speed, temperature and precipitation has been added in the supplementary information (Fig. S2) and is described in the methodology section. Moreover, the average wind directions and speed during each measurement loop are now reported in a wind rose plot in Fig. 4 and 5 (spatial distributions for Tartu and Tallinn, respectively) and are fully discussed in the manuscript.

As suggested by anonymous referees 2 and 4, a detailed analysis of the source apportionment diagnostics has been added in the revised manuscript. A figure including (a) $Q/Q_{exp}$ as a function of the number of factors, (b) correlations between OA sources with external factors as a function of the number of factors and the decrease in $Q/Q_{exp}$ time series (c) and profiles (d) for increasing number of factors has been added in the supplementary information (Fig S5). Moreover, a table reporting the correlations between the OA sources from our four-factor solution and literature profiles has been added in the main text.

Moreover, following the suggestion of anonymous referee 4, we have added the correlation coefficients ($R^2$) between the spatial distributions of all sources and compounds in Tartu in the revised manuscript (Table S1).

Lastly, in order to give an overview of the major local PM sources, we have added emission maps in the revised manuscript (Fig. S1). The wood combustion and industrial sources and the traffic emission rates of the main streets are reported in these maps.

**Anonymous Referee #1**

*This paper uses mobile measurements to spatially map aerosols and trace gases in two Estonian cities. The use of high time resolution instruments means that good spatial resolution is obtainable and the results are systematically analysed to present statistics on the urban increments. The use of source apportionment techniques on the AMS and Aethalometer data adds extra depth to the results. Overall, the analysis appears solid and the results are well presented. I recommend publication subject to minor (mostly technical) revisions.*

**Page 1, line 1:** *Please be more specific when referring to 'polluted continental areas'.*

**Changes in text:**
Page 1, Line 30: A strong increase in the secondary organic and inorganic components was observed during periods with transport of air masses from northern Germany, while the primary local emissions accumulated during periods with temperature inversions.

***Page 2, line 18:*** *PM2.5 has been the major focus for over 20 years now, so can hardly be described as 'recent'.*

**Author's response:** We fully agree with the reviewer's comment and have removed "recently" in the revised manuscript.

**Changes in text:**
Page 2, Line 21: Major attention has been devoted to the study of the $PM_{2.5}$ fraction (particulate matter with an aerodynamic equivalent diameter $d_{aero} \le 2.5$ µm), which has been linked…

***Page 6, line 20:*** *Remove the word 'highly'.*

**Author's response:** Removed in the revised manuscript.

***Page 11:*** *More explanation and justification of the P05 method should be given. Why was the 5th percentile chosen? What specific effect was the subtraction expected to achieve? Given that the base of the sigmoid fits was allowed to vary, why is this even necessary?*

**Author's response:** This is a good point raised by the reviewer that needs additional explanation. The subtraction of P05 is needed in order to decrease the background variability of each single loop before averaging. This step could be skipped if the sigmoid fit could be applied to single loops, but this is not possible due to the high variability in the signal within each single loop. We tested the sensitivity of the method by subtracting P10 instead of P05, and no major changes were observed in the results. This information has been added in the revised manuscript.

**Changes in text:**
Page 14 Line 7: In most  cases the base of the sigmoid function is slightly above zero. This indicates that the  subtracted P05 didn't represent the full regional background, which is therefore given by the sum of the average P05 and the base of the sigmoid function. Note that the initial subtraction of P05 would not be necessary if the longitudinal profile of each single loop could be fitted. However, this is not possible due to the high concentration variability within each single loop. A sensitivity analysis was performed by using P10 instead of P05 and no major changes were observed in the final results.

***Page 12, line 13:*** *The use of the median does not completely rule out the influence of the kerbside increment because street canyon effects (e.g. when the local wind is perpendicular to a road) can cause on-road emissions to persist is this microenvironment and the increment to no longer manifest itself as discrete spikes in the data. This would cause the median to increase over what would be expected of the urban background. Because these measurements are taken on-road, it is perhaps inevitable that estimates of the urban background will be biased slightly high because of the influence of traffic sources, so this should be added as a caveat. It may be possible to exclude this by selectively averaging the less-busy roads.*

**Author's response:** This has been discussed further in the revised manuscript.

**Changes in text:**
Page 13, Line 27: While the averaged profiles take into account the effects of the measured point sources in the urban area (mostly traffic and residential emissions), the use of the median profiles is expected to represent more selectively exclude these effects, making the results more representative of the urban background concentrations. We note that the influence of curbside increments may not be completely removed when using median increments (e.g. accumulation of traffic emissions due to street canyon effects), and therefore these increments might be biased high and should be regarded as our highest estimates of urban background concentrations.

*Page 12, line 3: Replace 'component' with 'components'.*

**Author's response:** Replaced in the revised manuscript

*Page 15: How was the temperature difference between 0 and 22m measured?*

**Author's response:** The temperature is measured at different heights in a meteorological tower at the Tallinn Zoo monitoring station.

---

## Author Comment (AC2) · 18 May 2016

**Author's response:**

We thank the reviewers for a careful reading and correction of our manuscript. Their suggestions have strongly helped improving the quality of the manuscript.

Following the suggestions of the anonymous referees 2, 3 and 4 we have added in the revised manuscript the description of the meteorological conditions during the measurement periods in both cities. A figure with the time series of wind direction and speed, temperature and precipitation has been added in the supplementary information (Fig. S2) and is described in the methodology section. Moreover, the average wind directions and speed during each measurement loop are now reported in a wind rose plot in Fig. 4 and 5 (spatial distributions for Tartu and Tallinn, respectively) and are fully discussed in the manuscript.

As suggested by anonymous referees 2 and 4, a detailed analysis of the source apportionment diagnostics has been added in the revised manuscript. A figure including (a) $Q/Q_{exp}$ as a function of the number of factors, (b) correlations between OA sources with external factors as a function of the number of factors and the decrease in $Q/Q_{exp}$ time series (c) and profiles (d) for increasing number of factors has been added in the supplementary information (Fig S5). Moreover, a table reporting the correlations between the OA sources from our four-factor solution and literature profiles has been added in the main text.

Moreover, following the suggestion of anonymous referee 4, we have added the correlation coefficients ($R^2$) between the spatial distributions of all sources and compounds in Tartu in the revised manuscript (Table S1).

Lastly, in order to give an overview of the major local PM sources, we have added emission maps in the revised manuscript (Fig. S1). The wood combustion and industrial sources and the traffic emission rates of the main streets are reported in these maps.

**Anonymous Referee #2**

**General Comments:**

*Elser et al. describe gas and aerosol measurements conducted in two Estonian cities. The authors use a mobile platform to investigate the extent to which pollution concentrations within the city limits exceed regional background levels. Via source apportionment, the authors attribute the observed organic aerosol loadings to four primary sources: traffic emissions, biomass burning, primary residential emissions, and secondary aerosol formation. The authors map the spatial distribution of each component to identify source "hot spots." In both cities, traffic-related components were most variable in the city center while biomass burning and primary residential emissions were concentrated in populated regions. Secondary components were well distributed throughout each city and had the least impact by point sources; however, increases in secondary aerosol were most strongly influenced by long-range transport of air masses originating from polluted regions west of Estonia.*

*The paper presents a useful and effective methodology for studying the impact of point sources on local air quality. Furthermore, these measurements are important as they assess the pollutant levels and source contributions in an understudied region of Europe. The manuscript is very well organized and many of the conclusions are well supported. I have some recommendations that would improve the quality of the manuscript. Upon addressing these comments, I recommend the manuscript for publication.*

*First, I believe that further details of the source apportionment should be included to strengthen the argument of a four-factor solution. Questions and comments pertaining to this aspect of the manuscript are summarized below. Second, parts of the methodology section employ sentence structure and wording that, at times, is awkward and/or difficult to follow. While I do not wish to interfere with stylistic choices, I believe that some rewording of these sections may help the manuscript read more fluently. Suggestions are provided in the Minor Comments.*

**Source Apportionment**

*The authors identify four factors that sufficiently describe the variation in the data. The authors are thorough with the comparison of factors with external tracers; however, there is little discussion and no figures demonstrating the model residuals as the PMF solution is pushed to higher factors. The authors describe their observations (Page 10, lines 12 – 24), however a figure should be included demonstrating the behavior of $Q/Q_{exp}$ as a function of the number of factors. Furthermore, the authors present a 5-factor solution, but argue that the fifth "unknown" factor exhibits a primary emission temporal pattern (which is uncharacteristic of a LV-OOA factor) and therefore does not significantly improve the interpretation of data. While this may be true, I believe it is necessary to demonstrate that the temporal residuals are not significantly improved for a 5-factor solution. It may be that the "unknown" factor results from factor splitting or some other mathematical construct. In any case, the PMF discussion should better describe the factor residuals.*

*The authors invoke bootstrapping as a means of constraining the error in the PMF solution. The author's note that bootstrapping inherently varies the algorithm starting point (i.e., seed) and therefore accounts for model uncertainties; however, the PMF solution may also be strongly affected by variations in fpeak (Ulbrich et al. 2009). There is little discussion about the rotational ambiguity of the PMF solution. I believe this discussion is necessary in order to evaluate the robustness of the PMF solution.*

*Finally, I believe it would be useful to compare the factor profiles to published spectra. This comparison would provide additional justification for the resolved factors. Specifically, I have some questions regarding the RIOA factor. The RIOA factor exhibits a temporal pattern that appears to be unique; however, the RIOA factor only exists in a 4 or higher factor solution and is primarily resolved from the BBOA and HOA factors (Fig S3). Consequently, the factor associated with RIOA results from the contribution of two other primary emission factors. While this may be simply due to the fact that RIOA, BBOA, and HOA are common in residential areas, this result may also be a result of factor splitting.*

*There are a number of ways the authors can provide additional evidence in support of the RIOA factor. The simplest option would be to compare the factor profiles to published spectra. The authors provide some comparison in the text, however a supplemental figure would be more illustrative. A more thorough analysis would be to perform PMF on subsets of the data and determine if the RIOA factor is still resolved. For example, if one were to remove time periods when the RIOA component is dominant, does PMF still resolve an RIOA factor? I believe these additional tests would strengthen the authors' PMF solution.*

**Author's response:**

Based on the reviewer comment, we have additionally performed a thorough residual analysis as a function of the number of factors. These diagnostics are presented in the new figure Fig. S5 in the supplementary of the revised manuscript (see below). This figure includes the change in $Q/Q_{exp}$, in the correlation coefficients ($R^2$) of the resolved factors with the external markers and the change in the residuals time series and profiles for solutions with increasing number of factors. We show that the correlation coefficients ($R^2$) between factors and markers increase when a fourth factor is included, but are not improved when a fifth factor is added. The addition of the fourth factor, which enabled the extraction of RIOA, allows explaining additional structures in the residuals' time series and unsaturated fragments in the residuals mass spectrum. Including a fifth factor also improves the model mathematical quality, by additionally explaining $C_xH_yN_w$ and biomass burning (at $m/z$ 60 and 73) related fragments. The additionally extracted factor in the five-factor solution, referred to as 'unknown', has elevated contributions from oxygenated fragments often related to SOA ($m/z$ 44) and BBOA ($m/z$ 60 and 73), but a time series that unambiguously relates this factor to a spatially variable primary emission source. In effect, the majority (62%) of this factor contribution arises from a split in the BBOA factor from the four-factor solution (the rest comes from the residuals and the OOA). Moreover, the sum of the contributions of the 'unknown' factor and the BBOA from the five-factor solution matches the BBOA contributions from the four-factor solution ($R^2 = 0.97$ and slope = 1.15 as shown in Fig. S6). This split in the BBOA is very likely a direct consequence of the variable nature of this combustion source, but the two BBOA-like factors extracted in the five-factor solution could not be related to different emission processes. The addition of this factor did not affect the spectral profiles and time series of the other factors and their correlations with their respective markers and did not aid the interpretation of the data. Therefore, we considered the four-factor solution as an optimal representation of our data. This discussion is now added in the text.

The $f_{peak}$ approach has been used in the past to study the rotational uncertainty of the source apportionment solution. However, varying the fpeak parameter allows to trace only one dimension through the rotationally accessible domain (which is multi-dimensional), and therefore provides only a lower limit for rotational uncertainty (Paatero et al., 2014). Bootstrap is a more effective approach to explore the rotational ambiguity and provides an upper limit for the rotational uncertainty.

A table containing the correlation coefficients ($R^2$) between the OA profiles from the four-factor solution and literature profiles has been added in the main text of the revised manuscript (Table 2). The high correlations retrieved for the RIOA with cooking spectra from literature ($R^2$ of about 0.8), strengthens the use of a four-factor solution and the link between our RIOA spectra and cooking emissions.

As described in the text, within the bootstrap method 64 % of the original points are used in each replicate of the input matrices. For all one-hundred bootstrap runs the RIOA was retrieved and the solution was very stable. Moreover, we also performed some PMF runs using only the data from Tallinn, where RIOA is more homogeneous compared to Tartu, and this factor was still always resolved. Therefore, this factor will always be resolved even when performing PMF on smaller subsets of the data.

**Changes in text:**

Page 10, Line 31: Some important diagnostic parameters of the source apportionment (including $Q/Q_{exp}$, factor-marker correlation, and time-series and profiles residuals for solutions with different number of factors) are reported in Fig. S5. The correlation coefficients

($R^2$) between factors and markers significantly increase when a fourth factor is included, but are not improved when a fifth factor is added. The addition of the fourth factor, which enabled the extraction of RIOA, allows explaining additional structures in the residuals' time series and unsaturated fragments in the residuals mass spectrum. Including a fifth factor also improves the model mathematical quality, by additionally explaining $C_xH_yN_w$ and biomass burning (at $m/z$ 60 and 73) related fragments. The additionally extracted factor in the five-factor solution, referred to as 'unknown', has elevated contributions from oxygenated fragments often related to SOA ($m/z$ 44) and BBOA ($m/z$ 60 and 73), but a time series that unambiguously relates this factor to a spatially variable primary emission source. In effect, the majority (62%) of this factor contribution arises from a split in the BBOA factor from the four-factor solution (the rest comes from the residuals and the OOA). Moreover, the sum of the contributions of the 'unknown' factor and the BBOA from the five-factor solution matches the BBOA contributions from the four-factor solution ($R^2$ = 0.97 and slope = 1.15 as shown in Fig. S6). This split in the BBOA is very likely a direct consequence of the variable nature of this combustion source, but the two BBOA-like factors extracted in the five-factor solution could not be related to different emission processes. Furthermore, the addition of this factor did not affect the spectral profiles and time series of the other factors and their correlations with their respective markers and did not aid the interpretation of the data. Therefore, we considered the four-factor solution as an optimal representation of our data. Table 2 contains the correlation coefficients ($R^2$) between the OA profiles from the four-factor solution and available literature profiles (Aiken et al., 2009; Mohr et al., 2012; Setyan et al., 2012; Crippa et al., 2013b). The high correlations obtained in all cases support the use of a four-factor solution and strengthen the link between the RIOA and cooking emissions ($R^2$ of about 0.8 between RIOA and cooking tracer).

~~If the number of factors is decreased, the RIOA factor is not resolved and the OOA time-series becomes contaminated by local spikes, which is unexpected for a regional component (see Fig. S3 and S4). In contrast, if a five-factor solution is considered an additional highly oxygenated factor is obtained ("unknown" factor in Fig. S3 and S4). The mass spectrum of this additional factor resembles a low-volatility OOA (LV-OOA), as resolved in many previous works (Jimenez et al., 2009), but its time series exhibits the typical characteristics of the primary factors, i.e. strong increases in emission areas. Therefore, this further increase in the number of factors doesn't seem to improve the interpretation of the data, as the new factor cannot be explicitly associated to distinct sources or processes. Accordingly, a four-factor solution was considered as optimal and is utilized below.~~

[Figure]

Figure S5 (new): Source apportionment diagnostics for increasing number of factors: (a) $Q/Q_{exp}$; (b) Correlation coefficient ($R^2$) between OA sources and markers; (c) Decrease in $Q/Q_{exp}$ time series; (d) Decrease in $Q/Q_{exp}$ profiles.

[Figure]

Figure S6 (new): Correlation between the BBOA time series from the four-factor solution and the sum of the BBOA and the 'unknown' time series in the five-factor solution.

Table 2: Correlation coefficients ($R^2$) between the OA profiles from the four-factor solution and literature profiles. Note: The different nomenclatures used in the literature for the different OOA factors have been homogenized to a semi-volatile OOA (SV-OOA) and a low-volatility OOA (LV-OOA).

| $R^2$ | Aiken et al., 2009 | Mohr et al., 2012 | Setyan et al., 2012 | Crippa et al., 2013b |
|---|---|---|---|---|
| HOA-HOA | 0.82 | 0.96 | 0.72 | 0.78 |
| BBOA-BBOA | 0.86 | 0.68 | --- | --- |
| RIOA-COA | --- | 0.83 | --- | 0.81 |
| OOA-SVOOA | 0.96 | 0.72 | 0.90 | 0.71 |
| OOA-LVOOA | 0.91 | 0.93 | 0.94 | 0.96 |

**Specific Comments**:

***Page 4, Lines 29-30:*** *Are there any sources that outline the spatial distribution of heating systems within the city? For example, can the authors comment on why BBOA emissions are higher and more dispersed in region (2) of Tallin (Fig 1) as opposed to region (7)?*

**Author's response:**
To give an overview of the local PM sources, we have added emission maps of the major sources (including residential wood combustion, industry and traffic) in the revised manuscript. These maps are reported in the new Fig. S1 (see below), where residential wood combustion sources are marked with green dots, industrial sources (mainly local boiler houses) with blue markers and main streets are colored based on the traffic emission rates. It is clear that the density of sources of residential wood combustion is much higher in the southern part of the driving route in Tallinn (region 2), which is in agreement with the higher BBOA concentrations observed in this area.

**Changes in text:**

Page 4, Line 30: The measurements took place from 10 to 17 March 2014 in Tartu and from 25 March to 1 April 2014 in Tallinn.  Emission maps including residential wood combustion and industrial sources and the traffic emission rates in the major streets of the two cities are reported in Fig. S1. The driving routes  were chosen in order to cover heavily trafficked roads, residential areas  and background sites with little local emissions.

Page 14, Line 31: BBOA is  strongly enhanced in the residential areas, consistent with the distribution of residential wood combustion sources shown in Fig. S1. The maximum BBOA enhancement is seen in the evening hours (15:00 to 21:00, LT) when domestic heating is more active.

Page 16, Line 5: BBOA (Fig. 5d) has higher contributions in the  residential areas, especially in region 2 of the driving route, where there is a very high density of residential wood combustion sources (see Fig. S1). Compared to Tartu, in Tallinn the spatial distribution of RIOA (Fig. 5e) is more homogeneous, with only slight enhancements in the residential area and in the city center.

[Figure]

Figure S1 (new): Emission maps for (a) Tartu and (b) Tallinn. Green dots indicate residential wood combustion sources, blue markers indicate industrial sources (mainly local boiler houses) and the color of the main streets represents the traffic emission rates.

***Page 8, Line 5:*** *What studies have used the eBC source apportionment method? Please provide references.*

**Author's response:** We thank the reviewer for the valuable remark. We have added the following references in the modified manuscript: Favez et al., (2010), Herich et al. (2011), Sciare et al. (2011) and Crilley et al. (2015).

**Changes in text:**
Page 8, Line 14: This method is described in detail in Sandradewi et al. (2008) and has been successfully applied at many locations across Europe (Favez et al., 2010; Herich et al., 2011, Sciare et al., 2011; Crilley et al., 2015).

***Page 12, Lines 24-26:*** *From what directions do emissions in Tartu/Tallin drain? It would seem to make most sense to take the upwind concentrations as your regional background. Perhaps a discussion of the topography and typical springtime meteorological conditions would help orient the reader to understand which airspaces reflect background conditions.*

**Author's response:**
As mentioned in the introduction of this review, we have included detailed wind direction and wind speed analyses for the measurement periods in both cities in the revised manuscript. The time series of the wind direction, wind speed and the wind roses showing the average wind direction and speed during each loop are reported for the two measurement locations in the revised manuscript (Fig. S2, Fig. 4b and Fig. 5b). During the mobile measurements, the wind was predominantly from the west in Tartu, and from west and east in Tallinn. The west winds observed during the drives in Tartu (with speeds between 1 and 2.6 m s$^{-1}$), don't seem to influence the background concentrations measured in the east side of the loop, as the base values obtained for the east side are always equal or lower than those obtained in the west (see Table 3). As the differences between the east and west base values (from the sigmoid fits) are in most cases minor, the west-east averages were used to calculate the urban increments concentrations.
In Tallinn, in order to identify possible processes influencing the spatial distributions of the measured pollutants for the two different wind patterns, the average spatial distributions were calculated for loops with west winds (7 loops) and east winds (16 loops), excluding drives during accumulation events. The results of these analyses are reported in the supplementary of the revised manuscript (Fig. S14 and S15) and show that, in general, the wind direction doesn't have a visible effect on the identified source areas and similar enhancements are found for both wind directions. A detailed analysis of the spatial distributions shows that BBOA, SO$_4$ and NO$_3$ show considerably higher enhancements for west winds, while HOA is more increased for east wind conditions. This difference is most probably related to the presence of west winds during the weekend (enhanced residential emissions) and east winds during the week-day measurements (enhanced traffic emissions).

**Changes in text:**
Page 5, Line 10: Meteorological data were recorded on a meteorological tower in Külitse (around 10 km south-east from Tartu) and in the Tartu and Tallinn-Zoo meteorological stations. The most relevant parameters (including wind direction and speed, temperature and precipitation) are reported in Fig. S2.

Page 14, Line 14: As shown by the wind rose in Fig. 4b, during the drives in Tartu the wind was predominantly from the west. However, the background concentrations measured at the east side of the loop don't seem to be affected by the transport of pollutants from the urban area, as the base values obtained for the east side are equal or lower than those from the

west side (see Table 3).  As these differences between the west and east fits are in most cases rather low,  we use the west-east averages of the base values to calculate the urban increments concentrations in Table 2.

Page 16, Line 14: Winds from west and east were observed during the mobile measurements in Tallinn (Fig. 5b). In order to identify possible processes influencing the spatial distributions of the measured pollutants for the two different wind patterns, the average spatial distributions were calculated for al loops with west wind (7 loops) and loops with east wind (16 loops, excluding drives during accumulation events). The results of these analyses are reported in the supplementary information (Fig. S14 and S15) and show that, in general, the wind direction didn't have an effect on the identified source areas and similar enhancements were found for both types of winds. A detailed analysis of these spatial distributions shows that BBOA, $SO_4$ and $NO_3$ are stronger enhanced during west winds, while HOA is more enhanced for east wind conditions. This difference is most probably related to the presence of west winds during the weekend (enhanced residential emissions) and east winds during the week-day measurements (enhanced traffic emissions).

[Figure]

Figure S2 (New): Meteorological conditions during measurements periods in Tartu (data from the Tartu monitoring station) and Tallinn (data from the Zoo monitoring station).

[Figure]

Figure 4 (Modified): (a) Driving route in Tartu: the red trace represents the GPS data, the yellow star the stationary measurements location and the blue dots the monitoring stations of the Estonian Environmental Research Institute (EERC); (b) Wind conditions during the mobile measurements in Tartu: red traces represent the wind direction and speed for the single loops and the average of all loops is represented in blue; (c to k) Average spatial distributions of all identified OA sources (panels c to f) and other measured components (panels g to k) in Tartu. The color scales represent enhancement over the background concentrations; the maximum of the color scales is fixed to the 75th percentile of the average enhancement of each component in panels g to k and to the highest 75th percentile among all OA sources in panels c to f. The sizes of the points represent the number of points that were averaged in each case.

[Figure]

Figure 5 (Modified): (a) Driving route in Tartu: the red trace represents the GPS data, the yellow star the stationary measurements location and the blue dots the monitoring stations of the Estonian Environmental Research Institute (EERC); (b) Wind conditions during the mobile measurements in Tartu: red traces represent the wind direction and speed for the single loops and the average of all loops is represented in blue; (c to k) Average spatial distributions of all identified OA sources (panels c to f) and other measured components (panels g to k) in Tallinn. The color scales represent enhancement over the background concentrations; the maxima of the color scales have been fixed to the 75th percentile of the average enhancement of each component in panels g to k and to the highest 75th percentile among all OA sources in panels c to f. The sizes of the points represent the number of points that have been averaged in each case (Note: less data available for CO).

[Figure]

Figure S14 (New): Left: Average spatial distributions of the OA sources in Tallinn for west and east winds. The color scale represents the enhancement over the background concentrations and the size the number of points that have been averaged in each case. The data related to special events was excluded for these analyses. Right: Distribution of the normalized differences between the east- and west-related spatial distributions for each compound. $X_0$ indicates the center of the gauss function used to fit each distribution.

[Figure]

Figure S15 (New): Left: Average spatial distributions of the inorganic components ($SO_4$, $NO_3$, $NH_4$ and Cl), eBC and $CO_2$ in Tallinn for west and east wind conditions. The color scales represent the enhancement over the background concentrations and the size the

number of points that have been averaged in each case. The data related to special was excluded for these analyses. Right: Distribution of the normalized differences between the east- and west-related spatial distributions for each compound. $X_0$ indicates the center of the gauss function used to fit each distribution.

***Page13, Lines 1-3:*** *Here, you state that BBOA is most enhanced during the evening hours, while on Page 11, Line 7 you state that higher loadings during the day are attributed to an increase primary sources (including BBOA). These statements tend to contradict each other. Please clarify.*

**Author's response:** The hours between 7:00 to 19:00, LT and 19:00 to 7:00, LT are referred to as day-time and night-time respectively, while we consider evening hours when biomass burning contribution is highest during the hours between 15:00 to 21:00, LT. So, there is an overlap between evening and day-time hours. This has been clarified in the revised manuscript.

***Page 13, Lines 1-13:*** *Here, you discuss diurnal patterns. If possible (perhaps with the stationary measurements), it would be most illustrative if the diurnal patterns were included as a figure.*

**Author's response:** We agree with the reviewer that the diurnal patterns would help in our analysis. However, such analyses were not possible with our data, as stationary measurements were performed mostly overnight and mobile measurements are strongly affected by point sources. Nevertheless, similar analyses were performed using mobile conditions only, upon averaging the data over longer time periods (i.e. 2h, see Fig. S13). The discussion in Lines 1-13 in Page 13 is derived from these analyses.

***Figures 5 and 6:*** *Consider adding the labeling from Fig 1 to these plots in order to facilitate the identification of source regions.*

**Author's response:** We have deleted Fig. 1 and have added the driving routes for Tartu and Tallinn in Fig. 4 and 5 of the revised manuscript.

*Minor Comments*

*The following are wording suggestions that may help improve the fluency of the methodology section*

**Author's response:** We thank the reviewer for the useful recommendations. The following suggestions have been introduced in the revised manuscript:

***Section 2.1***
***Page 4, Line 13:*** *"… Tartu, with an area of 38.8 km2…"*
***Page 4, Line 21:*** *"…to strongly enhance the signal of traffic emissions …"*
***Page 4, Line 23:*** *"… with low stacks in both cities. In this regard, a detailed…."*
***Section 2.2***
***Page 5, Line 17:*** *"…For this work , the AMS …"*
***Page 6, Line 1:*** *"… measurement method automatically corrects for the loading effect …"*
***Page 6, Lines 3-6:*** *" The concentrations of trace gases were measured by a Picarro-G2301 CO2/CH4 analyzer and a Licor-6262 CO monitor. "*
***Section 2.3***
***Page 6, Lines 15-16:*** *"…collection efficiency (CE) algorithm by Middlebrook et al. (2012) was used in the calculation of ambient mass concentrations (Middlebrook et al., 2012)."*
***Section 2.4***
***Page 4, Line 24:*** *"…allows the representation of a two-dimensional …"*
***Page 7, line 4:*** *"In our case, the model input are the data and error matrices of OA…"*
***Page7, Line 6:*** *"…contain the fits to the high-resolution data (292 ions)…"*
***Page 7, Line 7:*** *"…agrees with the mass calculated from the unit mass resolution integration…"*
***Page 7, Line 13:*** *"…directly calculated from the CO2+ fragment using the organic …"*
***Page 7, Line 14:*** *"… were excluded from the PMF analysis…"*
***Page 7, Line 15:*** *"… variability of the CO2+; these ions were reinserted post-analysis"*
***Page 7, Line 18:*** *"… replicate datasets resulting from the perturbation of the original data…"*
***Page 7, Line 20:*** *" … while other rows are removed (Paatero et al., 2014)…"*
***Page 7, Line 23:*** *"Note that as each bootstrap…"*
***Page 7, Line 24:*** *"… initialization point; thus, this methodology inherently includes the investigation of the classic seed variability…"*
***Page 7, Line 25:*** *"…consistent, suggesting that the solution is robust."*
***Section 2.4.2***
***Page 8, Line 8:*** *"… for the correlations with the external tracers, but their spatial distributions couldn't be explored…"*

***Page 5, Line 20:*** *"…lens efficiently transmits particles with 80 nm < Dp ≤ 3 µm and has been tested in previous chamber and ambient studies (Williams et al., 2013; Wolf et al., 2015; Elser et al., 2015)"*

**Author's response:** As mentioned in the text and described by Williams et al. (2013), the $PM_{2.5}$ lens efficiently transmits particles between 80 nm and up to at least 3 µm. The fact that particles larger than 3 µm could also be transmitted efficiently with this system is an important detail that needs to be considered in the presence of large particles. Therefore we prefer to keep this information in the text.

**References:**

Aiken, A. C., Salcedo, D., Cubison, M. J., Huffman, J. A., DeCarlo, P. F., Ulbrich, I. M., Docherty, K. S., Sueper, D., Kimmel, J. R., Worsnop, D. R., Trimborn, A., Northway, M., Stone, E. A., Schauer, J. J., Volkamer, R. M., Fortner, E., de Foy, B., Wang, J., Laskin, A., Shutthanandan, V., Zheng, J., Zhang, R., Gaffney, J., Marley, N. A., Paredes-Miranda, G., Arnott, W. P., Molina, L. T., Sosa, G., and Jimenez, J. L.: Mexico City aerosol analysis during MILAGRO using high resolution aerosol mass spectrometry at the urban supersite (T0) – Part 1: Fine particle composition and organic source apportionment, Atmos. Chem. Phys., 9, 6633-6653, 2009.

Crilley, L. R., Bloss, W. J., Yin, J., Beddows, D. C. S., Harrison, R. M., Allan, J. D., Young, D. E., Flynn, M., Williams, P., Zotter, P., Prevot, A. S. H., Heal, M. R., Barlow, J. F., Halios, C. H., Lee, J. D., Szidat, S., and Mohr, C.: Sources and contributions of wood smoke during winter in London: assessing local and regional influences, Atmos. Chem. Phys., 15, 3149–3171, 2015.

Crippa, M., El Haddad, I., Slowik, J. G., DeCarlo, P. F., Mohr, C., Heringa, M. F., Chirico, R., Marchand, N., Sciare, J., Baltensperger, U., and Prévôt A. S. H.: Identification of marine and continental aerosol sources in Paris using high resolution aerosol mass spectrometry, J. Geophys. Res., 118, 1950–1963, 2013b.

Favez, O., El Haddad, I., Piot, C., Boréave, A., Abidi, E., Marchand, N., Jaffrezo, J.-L., Besombes, J.-L., Personnaz, M.-B., Sciare, J., Wortham, H., George, C., and D'Anna, B.: Inter-comparison of source apportionment models for the estimation of wood burning aerosols during wintertime in an Alpine city (Grenoble, France), Atmos. Chem. Phys., 10, 5295–5314, 2010.

Herich, H., Hueglin, C., and Buchmann, B.: A 2.5 year's source apportionment study of black carbon from wood burning and fossil fuel combustion at urban and rural sites in Switzerland, Atmos. Meas. Tech., 4, 1409–1420, 2011.

Mohr, C., DeCarlo, P. F., Heringa, M. F., Chirico, R., Slowik, J. G., Richter, R., Reche, C., Alastuey, A., Querol, X., Seco, R., Peñuelas, J., Jiménez, J. L., Crippa, M., Zimmermann, R., Baltensperger, U. and Prévôt, A. S. H.: Identification and quantification of organic aerosol from cooking and other sources in Barcelona using aerosol mass spectrometer data, Atmos. Chem. Phys., 12, 1649–1665, 2012.

Paatero, P., Eberly, S., Brown, S. G., and Norris, G. A.: Methods for estimating uncertainty in factor analytic solutions, Atmos. Meas. Tech., 7, 781–797, 2014.

Sciare, J., d' Argouges, O., Sarda-Estève, R., Gaimoz, C., Dolgorouky, C., Bonnaire, N., Favez, O., Bonsang, B., and Gros, V.: Large contribution of water-insoluble secondary organic aerosols in the region of Paris (France) during wintertime, J. Geophys. Res. Atmos., 116, D22203, 2011.

Setyan, A., Zhang, Q., Merkel, M., Knighton, W. B., Sun, Y., Song, C., Shilling, J. E., Onasch, T. B., Herndon, S. C., Worsnop, D. R., Fast, J. D., Zaveri, R. A., Berg, L. K., Wiedensohler, A., Flowers, B. A., Dubey, M. K., and Subramanian, R.: Characterization of submicron particles influenced by mixed biogenic and anthropogenic emissions using high-resolution aerosol mass spectrometry: results from CARES, Atmos. Chem. Phys., 12, 8131-8156, 2012.

---

## Author Comment (AC3) · 18 May 2016

**Author's response:**

We thank the reviewers for a careful reading and correction of our manuscript. Their suggestions have strongly helped improving the quality of the manuscript.

Following the suggestions of the anonymous referees 2, 3 and 4 we have added in the revised manuscript the description of the meteorological conditions during the measurement periods in both cities. A figure with the time series of wind direction and speed, temperature and precipitation has been added in the supplementary information (Fig. S2) and is described in the methodology section. Moreover, the average wind directions and speed during each measurement loop are now reported in a wind rose plot in Fig. 4 and 5 (spatial distributions for Tartu and Tallinn, respectively) and are fully discussed in the manuscript.

As suggested by anonymous referees 2 and 4, a detailed analysis of the source apportionment diagnostics has been added in the revised manuscript. A figure including (a) $Q/Q_{exp}$ as a function of the number of factors, (b) correlations between OA sources with external factors as a function of the number of factors and the decrease in $Q/Q_{exp}$ time series (c) and profiles (d) for increasing number of factors has been added in the supplementary information (Fig S5). Moreover, a table reporting the correlations between the OA sources from our four-factor solution and literature profiles has been added in the main text.

Moreover, following the suggestion of anonymous referee 4, we have added the correlation coefficients ($R^2$) between the spatial distributions of all sources and compounds in Tartu in the revised manuscript (Table S1).

Lastly, in order to give an overview of the major local PM sources, we have added emission maps in the revised manuscript (Fig. S1). The wood combustion and industrial sources and the traffic emission rates of the main streets are reported in these maps.

**Anonymous Referee #3**

*The paper by Else et al. summarizes mobile springtime measurements of aerosol concentrations and several gas phase species in two Estonian cities. The measurements allowed the authors to identify 4 classes of OA in both cities. Overall, aerosol composition in both cities was similar and was dominated by higher concentration of primary types of OA during the day and by lower amount of secondary OA at night. Contribution of the secondary inorganic species was low except during a transport event.*
*The manuscript is very well-organized and well-written. There are two aspects that need some work in my opinion. One is related to wind direction and its variability during day and night sampling and how it might affect the interpretation of the results (see my comment below). The other aspect is that since the measurement was done in two cities, I think more can be done to compare quantitatively aerosol air quality in these two cities. Since measurements of CO are already available, I think it will be valuable to look at the enhancement ratios (not by subtracting a background) but considering scattering plots of say OA vs. CO, BC vs CO (or the PMF-resolved factors or other species vs. CO) in comparable times of the day to separate out the differences in dynamics, boundary layer heights, dilution, etc. and be able to determine a more valuable comparison of the aerosol sources in these cities. This will also allow the authors to compare the measurements with other*

*measurements (ground based on airborne) in other cities around the world. I support publishing the paper after my comments (above and below) are addressed.*

**Author's response:**
Indeed, ratios of different OA components to CO are generally used to take into consideration the effect of the PBL and dilution in order to investigate the evolution of a plume with photochemistry. However, here we lack measurements to estimate the photochemical lifetime of the sampled air masses (e.g NOx/NOy or VOCs).

In the figures below we display the scatter plots of different aerosol components versus CO to investigate differences between the two locations and with the time of the day. Such plots mostly reflect the profiles of combustion sources that dominate CO emissions. Similar ratios are found for the two cities (Fig. R1), which is consistent with similar sources of CO and similar emission profiles at both locations. HOA and eBC show higher ratios to CO during daytime (Fig. R2), as traffic (which dominates eBC emissions) is more enhanced during day-time compared to CO, which can be also affected by other sources (i.e. BBOA and RIOA).

[Figure]

Figure R1: Scatter plots of all particle phase components versus CO, color-coded by measurement location.

[Figure]

Figure R2: Scatter plots of all particle phase components versus CO, color-coded by measurement time.

*Abstract: indicate which month/season the measurements were carried out.*

**Author's response:** This information has been added in the abstract of the revised manuscript.

**Changes in text:**
Page 1, Line 14: This work presents the first spatially-resolved detailed characterization of the PM$_{2.5}$ in two major Estonian cities, (Tallinn and Tartu), using mobile measurements. The measurements were performed in March 2014 using a mobile platform. In both cities, the non-refractory (NR)-PM$_{2.5}$ was characterized by a high-resolution time-of-flight aerosol mass spectrometer (HR-ToF-AMS) using a recently developed lens which increases the transmission of super-micron particles.

*P5, L3: It is mentioned that stationary measurements were made at night. Were there any mobile measurements also carried out at night?*

**Author's response:** The mobile measurements were performed between 6:00 and 23:00 LT, including only few drives during the early morning and at nighttime.

*P8, L2: Explain why A(abs)=1.7 was used for wood burning BC? And why was it that the lower wavelength of 370 nm was not used? Doesn't it make sense to use 370 nm since BrC would be stronger there?*

**Author's response:** The choice of the wavelengths and of the angstrom exponents used in this work are based on the findings in Zotter et al. (In prep.), where radiocarbon ($^{14}$C) measurements of elemental carbon (EC) are combined with Aethalometer data to determine the best Angstrom exponents for wood burning ($\alpha_{WB}$) and traffic ($\alpha_{TR}$). The best α values were evaluated by fitting the source apportionment results of the Aethalometer (in particular $BC_{tr}$/BC) against the fossil fraction of EC ($EC_f$/EC) derived from $^{14}$C measurements. This analysis resulted in $\alpha_{tr}$ = 0.9 and $\alpha_{wb}$ = 1.68 fitting best the data when using the attenuation measured at 470 and 950 nm. Other wavelength combinations were also tested but in all cases, especially when 370 nm was used, the residuals of the fit were worse. Moreover it is known that the 370 nm channel of the Aethalometer is more sensitive to artefacts, including response to light absorbing SOA and the adsorption of VOCs on the filter. A brief description of the Aethalometer source apportionment method and the findings in Zotter et al. (In prep.) has been added in the revised manuscript.

**Changes in text:**
Page 8, Line 9: The Aethalometer measurements can be used to separate eBC from wood burning ($eBC_{wb}$) and from traffic ($eBC_{tr}$), by taking advantage of the spectral dependence of absorption, as described by the Ångström exponent (Ångström, 1929). Specifically, the enhanced absorption of wood burning particles in the ultraviolet and visible wavelengths region (370–520 nm) relative to that of traffic particles is used to separate the contributions of the two fractions. This method is described in detail….

Page 8, Line 19: The absorption Ångström exponent was calculated using the absorption measured at 470 and 950 nm and Ångström exponents of 0.9 and 1.7 were used for traffic and wood burning, respectively. These parameters were chosen following the suggestions in Zotter et al. (In prep.), where the comparison between radiocarbon ($^{14}$C) measurements of elemental carbon (EC) and the Aethalometer source apportionment results allowed the identification of the best wavelengths and Ångström exponents pairs.

*P8, L16-17: Just looking at Figure 2, it seems standard deviations of the averages would be really high, and maybe that's why they're not indicated along with the average values in Panel B. I wonder if estimates of the median values (or to be more complete, box and whisker plots) of the tracers will be more valuable than the average values.*

**Author's response:** We decided not to report the standard deviations in Fig. 2b (now Fig. 1b) as the big variability of the data can be seen from the time series and in our case it's simply reflecting the driving conditions. Thus, we believe that the standard deviation of the time series would not provide any useful information in our case, especially because this variability can be explained by the spatial distribution of the sources. Moreover, the addition of standard deviations of the time series could introduce some confusion between the meanings of the errors in different cases, as for the source apportionment results (see Fig. 3 of the revised manuscript) we report standard deviations among the 100 bootstrap runs (which is an indication of the model uncertainty and not of the temporal variability of the sources).

***P11, L15:*** *missing a word ": : :of ?? (data??)..."*

**Author's response:** We have modified this paragraph in the revised manuscript.

**Changes in text:**
Page 12, Line 26: RIOA is also enhanced during day-time in Tartu (27% compared to 20% during night-time), and has similar relative contributions for day- and night-time in Tallinn (20 and 22%, respectively). In contrast, BBOA shows similar relative contributions for day- and night-time in Tartu ( representing about 17 % of the OA mass), and slightly lower contributions during  day-time in Tallinn (20 % during day-time and 25 % at night-time).

***P12, L1:*** *replace kurbside with curbside*

**Author's response:** Replaced in the revised manuscript.

***Figure 2:*** *I suggest having the inorganics on a separate axis, with max _10 ug/m3, so you can see the tracers better.*

**Author's response:** The plot has been modified accordingly in the revised manuscript.

***Figure 5:*** *For some species, it appears that the conc. were very different on different sides of the loop, suggesting that the sources are towards the center of the loop (as opposed to one side, e.g., BBOA and sulfate). In other words there is gradient in the latitudinal direction as well as longitudinal direction. To further investigate the source regions, it makes sense to consider wind direction data with these distribution maps. Were wind directions consistent during the day and night sampling time? It seems the averages include both daytime and nighttime. Could you add average wind barbs representatives for daytime and nighttime or at least discuss the wind patterns in the text? Correct interpretation of the mean and median values in Table 2 with relation to the source regions also needs some knowledge of the wind direction.*

**Author's response:**
As mentioned above, we have added a time series with the meteorological parameters during the measurement periods (including wind direction and speed, temperature and precipitation) in the supplementary information of the revised manuscript. No systematical difference can be observed between day and night wind conditions.
In Tartu, during the mobile measurements west winds (with speeds between 1 and 2.6 m s$^{-1}$) were predominant (see wind rose in Fig. 4 in the revised manuscript). However, the wind doesn't seem to influence the background concentrations measured in the east side of the loop, as the base values obtained for the this side of the loop were always equal or lower than those obtained in the west (see Table 3). Therefore, we also exclude a big influence of the transport of pollutants within the urban area and we expect that the identified source areas will not be strongly biased by this effect.

***Figure 7:*** *Indicate in the legend that enhancement is relative to P05 values.*

**Author's response:** The Figure has been modified in the revised manuscript to indicate that the enhancement is relative to the P05 values. This information has also been added in the figure caption for further clarification.

**Changes in text:**
Figure 7 caption:  Average longitude profiles of the enhancements (above P05) of all measured components and sources in Tartu.

*Table 2 legend: Indicate which city the stats refer to.*

**Author's response:** Added in the revised manuscript.

**Changes in text:**
       (A) Average longitude profiles (Tartu):
       (B) Median longitude profiles (Tartu):

---

## Author Comment (AC4) · 18 May 2016

**Author's response:**

We thank the reviewers for a careful reading and correction of our manuscript. Their suggestions have strongly helped improving the quality of the manuscript.

Following the suggestions of the anonymous referees 2, 3 and 4 we have added in the revised manuscript the description of the meteorological conditions during the measurement periods in both cities. A figure with the time series of wind direction and speed, temperature and precipitation has been added in the supplementary information (Fig. S2) and is described in the methodology section. Moreover, the average wind directions and speed during each measurement loop are now reported in a wind rose plot in Fig. 4 and 5 (spatial distributions for Tartu and Tallinn, respectively) and are fully discussed in the manuscript.

As suggested by anonymous referees 2 and 4, a detailed analysis of the source apportionment diagnostics has been added in the revised manuscript. A figure including (a) $Q/Q_{exp}$ as a function of the number of factors, (b) correlations between OA sources with external factors as a function of the number of factors and the decrease in $Q/Q_{exp}$ time series (c) and profiles (d) for increasing number of factors has been added in the supplementary information (Fig S5). Moreover, a table reporting the correlations between the OA sources from our four-factor solution and literature profiles has been added in the main text.

Moreover, following the suggestion of anonymous referee 4, we have added the correlation coefficients ($R^2$) between the spatial distributions of all sources and compounds in Tartu in the revised manuscript (Table S1).

Lastly, in order to give an overview of the major local PM sources, we have added emission maps in the revised manuscript (Fig. S1). The wood combustion and industrial sources and the traffic emission rates of the main streets are reported in these maps.

**Anonymous Referee #4**

**General comments**

*This paper presents mobile measurements of gas- and particle-phase pollutants at two Estonian cities of Tartu and Talinn. Detail chemical composition of NR-PM2.5 as well as BC and trace gases (CO, CO2, and CH4) were observed in high time resolution and OA characteristics were found to be similar (HOA, BBOA, RIOA, and OOA) at both cities. Primary types of OA (HOA, BBOA and RIOA) were high during the day, whereas secondary OA (OOA) enhanced at night. In summary, the mobile measurements allowed the authors to observe time and spatial distribution of primary and secondary pollutants, as well as influences of long-range transport and local temperature inversion to aerosol and gases pollutants in Tartu and Talinn.*

*The manuscript is well structured and written. I have some comments regarding research methods, results and discussion as well as suggestions that would improve quality of the manuscript. There are also some typing errors in the text, tables and figures summarized in*

*technical comments. Overall, I support publication of the manuscript after my comments are addressed.*

**Specific comments**

**OA source apportionment:**
*I believe the authors have done rigorous analysis in selecting the best PMF factor solution. I expect some of the analysis figures and/or tables are provided in the SI. PMF diagnostic plots, such as those presented by Zhang et al. (2011), are important for understanding the analysis and discussion. Residuals of time series and mass spectra for 3-, 4-, and 5-factor solutions and correlations between factors time series and external tracers (i.e., CO, eBC, SO4, NO3) are useful in understanding selection of the best factor solution. I recommend adding this information in the SI at the least.*

**Author's response:**
As discussed above, a figure with the major source apportionment diagnostic plots has been added in the supplementary information (Fig. S5 in the revised manuscript). This figure includes the $Q/Q_{exp}$ as a function of the number of factors, the correlation coefficients ($R^2$) of the resolved factors with the external markers for solutions with different number of factors and the change in the residuals time series and profiles for increasing number of factors.

**Changes in text:** See complete changes in text in comment below regarding the five-factor solution.

**eBC source apportionment:**
*It is not clear why do the authors choose to calculate Angstrom exponent using absorption at 470 and 950 nm. Since Zotter et al. paper has not yet published, I could not verify how the calculation was done. I suggest adding a brief description of this calculation in the main text or SI. Also, brief description of calculation of eBCwb and eBCtr would be a useful addition in the SI.*

**Author's response:** The choice of the wavelengths and of the angstrom exponents used in this work are based on the findings in Zotter et al. (In prep.), where radiocarbon ($^{14}$C) measurements of elemental carbon (EC) are combined with Aethalometer data to determine the best Angstrom exponents for wood burning ($\alpha_{WB}$) and traffic ($\alpha_{TR}$). The best α values were evaluated by fitting the source apportionment results of the Aethalometer (in particular $BC_{tr}/BC$) against the fossil fraction of EC ($EC_f/EC$) derived from $^{14}$C measurements. This analysis resulted in $\alpha_{tr} = 0.9$ and $\alpha_{wb} = 1.68$ fitting best the data when using the attenuation measured at 470 and 950 nm. Other wavelength combinations were also tested but in all cases, especially when 370 nm was used, the residuals of the fit were worse. Moreover it is known that the 370 nm channel of the Aethalometer is more sensitive to artefacts, including response to light absorbing SOA and the adsorption of VOCs on the filter. A brief description of the Aethalometer source apportionment method and the findings in Zotter et al. (In prep.) has been added in the revised manuscript.

**Changes in text:**
Page 8, Line 9: The Aethalometer measurements can be used to separate eBC from wood burning ($eBC_{wb}$) and from traffic ($eBC_{tr}$), by taking advantage of the spectral dependence of absorption, as described by the Ångström exponent (Ångström, 1929). Specifically, the enhanced absorption of wood burning particles in the ultraviolet and visible wavelengths

region (370–520 nm) relative to that of traffic particles is used to separate the contributions of the two fractions. This method is described in detail….

Page 8, Line 19: The absorption Ångström exponent was calculated using the absorption measured at 470 and 950 nm and Ångström exponents of 0.9 and 1.7 were used for traffic and wood burning, respectively. These parameters were chosen following the suggestions in Zotter et al. (In prep.), where the comparison between radiocarbon ($^{14}C$) measurements of elemental carbon (EC) and the Aethalometer source apportionment results allowed the identification of the best wavelengths and Ångström exponents pairs.

**Results and discussion:**

**Page 10 Ln 1-2:** *Could the authors provide approximate uncertainty value of ratio of BBOA and eBCwb relative to change in Angstrom exponent?*

**Author's response:** As mentioned in the manuscript, the BBOA to $eBC_{wb}$ ratio is very sensitive to the chosen Ångström exponent for traffic. Zotter et al. (in prep.) recommend the use of Ångström exponents for traffic between 0.9 and 1.0, as using 1.1 leads to increased residuals in their method. Therefore, we used an Ångström exponent for traffic of 0.9 and reported the change in the BBOA to eBCwb ratio if an Ångström exponent for traffic of 1.0 had been used. This variability (ratio changed from 4 to 4.8) is an indication of the method uncertainty relative to the choice of Angstrom exponents, which is around 20%.

**Page 10 Ln 2-10:** *From information provided, it seems reasonable to assign RIOA as COA. The lack of diurnal variability does not mean that the factor is not a COA factor. It is possible that the lack of diurnal variability is due to homogeneous source of cooking emission, or stagnant atmosphere during measurements Have the authors look at meteorological conditions, such as wind direction and speed, when RIOA concentrations were high? I would recommend adding meteorological information in the SI.*
*Identification of RIOA or COA could be further assessed by plotting the factor mass spectra side-by-side with reference mass spectra from previous studies, such as by Mohr et al. (2012). The authors will need to provide more evidence to support identification of the RIOA factor.*

**Author's response:**
As previously mentioned, the diagnostic plots included in the revised manuscript support the use of a four-factor solution, where the RIOA factor is resolved. In general, an increase in the correlation coefficient ($R^2$) is observed when going from a three- to a four-factor solution. As the correlations are not further improved when considering higher order solutions, the four-factor solution is considered the best representation of the data. Also the trend in the model residuals supports the presence of four factors, as the decrease in the residuals time series is important when increasing from three to four factors but is rather low when further increasing the number of factors.

As mentioned above, we have added a time series with the meteorological parameters during the measurement periods (including wind direction and speed, temperature and precipitation) in the supplementary information of the revised manuscript (Fig. S2). We show that in general the wind direction does not influence our measurements significantly (see Fig. 4b and Fig. S14), but the temporal variability of the factor time series, including that of RIOA, are rather driven by their spatial distribution.

A table containing the correlation coefficients ($R^2$) between the OA profiles from the four-factor solution and literature profiles has been added in the main text of the revised manuscript (New table 2). The high correlation between RIOA and published cooking mass spectra suggests that RIOA may be heavily influenced by cooking processes. However, we could not exclude the contribution from other residential sources (e.g. waste or coal combustion), especially also due to the lack of statistically robust diurnal patterns for cooking that are not affected by the drives. Therefore, we prefer to refer to this factor to RIOA, rather than cooking.

**Changes in text:** See complete changes in text in the following comment.

*Page 10 Ln 16-24: The 5th factor, LV-OOA, may not show certain process or source, but it can show whether highly oxidized OA is important in the study locations. The LV-OOA can be formed from oxidation of primary OA (e.g., BBOA) or transported into the study location. The authors can discuss this further.*

**Author's response:**
We now added a thorough analysis for justifying the number of factors selected. We show that the addition of a fifth factor better explains $C_xH_yN_w$ and biomass burning (at $m/z$ 60 and 73) related fragments. The additionally extracted factor in the five-factor solution, referred to as 'unknown', has elevated contributions from oxygenated fragments often related to SOA ($m/z$ 44) and BBOA ($m/z$ 60 and 73), but a time series that unambiguously relates this factor to a spatially variable primary emission source. In effect, the majority (62%) of this factor contribution arises from a split in the BBOA of the four-factor solution (the rest comes from the residuals and the OOA). Consequently, the sum of the contributions of the 'unknown' factor and the BBOA from the five-factor solution matches the BBOA contributions from the four-factor solution ($R^2 = 0.97$ and slope = 1.15 as shown in Fig. S6). This split in the BBOA is very likely a direct consequence of the variable nature of this combustion source or may potentially represent its very rapid/immediate aging. That is, as the concentrations of this more oxygenated BBOA are highly variable in time and space and coincide with high BBOA concentrations, this factor cannot be transported into the study location, but rather represents an emission from another combustion regime or, less likely, an immediately transformed primary aerosol. As this factor could not be attributed to a specific process and its addition did not significantly alter the contribution from other sources, we have considered the four-factor solution as an optimal representation of our data. This discussion is now added in the text.

**Changes in text:**
Page 10, Line 31: Some important diagnostic parameters of the source apportionment (including $Q/Q_{exp}$, factor-marker correlation, and time-series and profiles residuals for solutions with different number of factors) are reported in Fig. S5. The correlation coefficients ($R^2$) between factors and markers significantly increase when a fourth factor is included, but are not improved when a fifth factor is added. The addition of the fourth factor, which enabled the extraction of RIOA, allows explaining additional structures in the residuals' time series and unsaturated fragments in the residuals mass spectrum. Including a fifth factor also improves the model mathematical quality, by additionally explaining $C_xH_yN_w$ and biomass burning (at $m/z$ 60 and 73) related fragments. The additionally extracted factor in the five-factor solution, referred to as 'unknown', has elevated contributions from oxygenated fragments often related to SOA ($m/z$ 44) and BBOA ($m/z$ 60 and 73), but a time series that unambiguously relates this factor to a spatially variable primary emission source. In effect, the majority (62%) of this factor contribution arises from a split in the BBOA factor from the four-factor solution (the rest comes from the residuals and the OOA). Moreover, the sum of

the contributions of the 'unknown' factor and the BBOA from the five-factor solution matches the BBOA contributions from the four-factor solution ($R^2$ = 0.97 and slope = 1.15 as shown in Fig. S6). This split in the BBOA is very likely a direct consequence of the variable nature of this combustion source, but the two BBOA-like factors extracted in the five-factor solution could not be related to different emission processes. Furthermore, the addition of this factor did not affect the spectral profiles and time series of the other factors and their correlations with their respective markers and did not aid the interpretation of the data. Therefore, we considered the four-factor solution as an optimal representation of our data. Table 2 contains the correlation coefficients ($R^2$) between the OA profiles from the four-factor solution and available literature profiles (Aiken et al., 2009; Mohr et al., 2012; Setyan et al., 2012; Crippa et al., 2013b). The high correlations obtained in all cases support the use of a four-factor solution and strengthen the link between the RIOA and cooking emissions ($R^2$ of about 0.8 between RIOA and cooking tracer).

~~If the number of factors is decreased, the RIOA factor is not resolved and the OOA time-series becomes contaminated by local spikes, which is unexpected for a regional component (see Fig. S3 and S4). In contrast, if a five-factor solution is considered an additional highly oxygenated factor is obtained ("unknown" factor in Fig. S3 and S4). The mass spectrum of this additional factor resembles a low-volatility OOA (LV-OOA), as resolved in many previous works (Jimenez et al., 2009), but its time series exhibits the typical characteristics of the primary factors, i.e. strong increases in emission areas. Therefore, this further increase in the number of factors doesn't seem to improve the interpretation of the data, as the new factor cannot be explicitly associated to distinct sources or processes. Accordingly, a four-factor solution was considered as optimal and is utilized below.~~

**Page 11 Ln 12-15:** *Increase of RIOA in Talinn is relatively small. I think distribution of RIOA in Talinn is more homogeneous compared to Tartu. Thus, enhancement of RIOA in urban area of Talinn is not well supported.*

**Author's response:** We thank the reviewer for the important remark. We have accordingly modified this paragraph in the revised manuscript.

**Changes in text:**
Page 12, Line 19: In terms of relative contribution, OOA is dominant during night-time, explaining on average between 42 and 44 % of the OA mass in Tartu and Tallinn, respectively.  The relative contribution of HOA to total OA mass is higher during day-time (32% in Tartu and 27% in Tallinn) than during night-time (20% in Tartu and 11% in Tallinn). RIOA is also enhanced during day-time in Tartu (27% compared to 20% during night-time), and has similar relative contributions for day- and night-time in Tallinn (20 and 22%, respectively). In contrast, BBOA shows similar relative contributions for day- and night-time in Tartu ( representing about 17 % of the OA mass), and slightly lower contribution during  day-time in Tallinn (20 % during day-time and 25 % at night-time).

**Page 13 Ln 1-3:** *Spatial distribution of eBC, CO, and CO2 are consistent not only with HOA but also with BBOA. Thus, they may come from BBOA as well. It would be easier to show consistency or inconsistency by correlation coefficient ($R^2$) between those tracers and HOA and BBOA.*

*Also, in general I disagree that CO2 is mostly traffic because it can be emitted from vegetation and other sources. The authors will need to provide more evidence to support CO2 from traffic.*

**Author's response:** We agree that $CO_2$ sources other than traffic are also present in urban areas (e.g. vegetation, BBOA…). However, our results indicate that these sources don't have a visible effect on the $CO_2$ enhancements in the urban area, as the spatial distribution of the $CO_2$ enhancement corresponds best to the one of HOA. These additional sources will indeed have an effect on the $CO_2$ background concentrations, which are subtracted for the calculation of the enhancements.

Following the suggestion of the referee, we have added a table in the supplementary information of the revised manuscript that contains all the correlation coefficients ($R^2$) between the spatial distributions of all sources and components in Tartu. This table confirms that the spatial distributions of the enhancements of eBC, $CO_2$ and CO are in good agreement with those of HOA ($R^2$ of around 0.6 in all cases), but no correlation is found between these components and BBOA ($R^2$ of 0.1 or lower).

**Changes in text:**
Page 13, Line 16: Lastly, the sizes of the points represent the number of measurement points that were averaged in each case. The correlation coefficients ($R^2$) between the spatial distributions of all sources and components are reported in Table S1.

Page 14, Line 29: The spatial distributions of the eBC, $CO_2$ and CO (Fig. 4i-k 5g-i) are consistent with that of HOA ($R^2$ of 0.61, 0.59 and 0.58, respectively), which indicates that these species originate mostly from traffic.

Table S1: Correlation coefficient ($R^2$) between the spatial distributions of all sources and components.

Tartu

| $R^2$ | HOA | BBOA | RIOA | OOA | $SO_4$ | $NO_3$ | $NH_4$ | Cl | eBC | $CO_2$ | CO | $CH_4$ |
|---|---|---|---|---|---|---|---|---|---|---|---|---|
| HOA | | 0.02 | 0.32 | 0.02 | 0.04 | 0.09 | 0.16 | 0.10 | **0.61** | **0.59** | **0.58** | 0.06 |
| BBOA | | | 0.47 | 0.05 | 0.28 | 0.20 | 0.16 | 0.47 | 0.03 | 0.01 | 0.10 | 0.11 |
| RIOA | | | | 0.08 | 0.35 | 0.14 | 0.22 | 0.39 | 0.24 | 0.17 | 0.28 | 0.07 |
| OOA | | | | | 0.12 | 0.09 | 0.12 | 0.21 | 0.01 | <0.01 | <0.01 | 0.01 |
| $SO_4$ | | | | | | 0.07 | 0.10 | 0.12 | 0.03 | 0.02 | 0.08 | 0.03 |
| $NO_3$ | | | | | | | **0.60** | 0.28 | 0.08 | 0.08 | 0.11 | 0.11 |
| $NH_4$ | | | | | | | | 0.24 | 0.09 | 0.05 | 0.11 | 0.10 |
| Cl | | | | | | | | | 0.07 | 0.03 | 0.11 | 0.13 |
| eBC | | | | | | | | | | **0.77** | **0.75** | 0.08 |
| $CO_2$ | | | | | | | | | | | **0.78** | 0.08 |
| CO | | | | | | | | | | | | 0.14 |
| $CH_4$ | | | | | | | | | | | | |

**Page 4 Ln 22:** _Add reference for these statements._

**Author's response:** Added in the revised manuscript.

**Page 5 Ln 10:** _Unit for flow rate is m3 s-1 or L min-1_

**Author's response:** Replaced "flow" by "velocity" in the revised manuscript.

**Page 5 Ln 12:** _What is the size of particle in the aerosol inlet before particles are divided into different aerosol measurements._

**Author's response:** There is no particle size segregation in the inlet of the mobile laboratory. Thus, the size of the measured particles depends on the instrument and inlet system cut-off. The inlet size cut-off is estimated to be at around 5 µm and the AMS lens at 2.5 µm. This information has been added in the revised manuscript.

**Changes in text:**
Page 5, Line 18: Two different inlet lines connected the main inlet to the aerosol and gas-phase instrumentation. The size cut-off of the inlet system was estimated to be around 5 µm.

**Page 5 Ln 28:** _eBC has been defined in the introduction._

**Author's response:** Corrected in the revised manuscript.

**Page 8 Ln 28-30:** _The statement about enhancement of negative health impacts is not well supported, as it was not within the scope of this study. I suggest the authors to omit the part or revise the sentence._

**Author's response:** We agree with the reviewer and have revised the sentence in the revised manuscript.

**Changes in text:**
Page 9, Line 11: Such intermittent pollution plumes (expected in some areas in a city) cannot be detected from stationary measurements at an urban background site, but  may be associated with negative health impacts.

**Page 10 Ln 29:** _Delete "secondary" or change it to "secondary source (OOA)"_

**Author's response:** Changed in the revised manuscript.

***Page 11 Ln 24:*** *"the 5th percentile (P05) of"*

**Author's response:** Corrected in the revised manuscript.

***Page 13 Ln 20:*** *Add space "… is 4.2 …"*

**Author's response:** Added in the revised manuscript.

***Table 2:*** *Superscript for the unit µg m-3.*

**Author's response:** Corrected in the revised manuscript.

***Figure 2:***
*(a) What does the different shade of purple for CO2 mean?*
*(b) If the average pollutant concentrations exclude those from special events, this needs to be included in the title.*

**Author's response:** (a) The different colors used for the $CO_2$ time series indicate data from different analyzers. Specifically, the light purple indicates data from the Licor analyzer, which is used in the period in which the Picarro analyzer was malfunctioning. This has been clarified in the figure legend of the revised manuscript.
(b) Yes, the special events were excluded for these calculations. This has been added in the figure caption of the revised manuscript.

***Figure 3:*** *I think mass spectra relative contribution is not in %. For comparison, relative contribution in Figure S3 is unitless.*

**Author's response:** We apologize for this mistake; the unit has been removed in the revised manuscript.

***Figure 8:*** *Add in the title that the back-trajectories is from 10:00 at 10 March 2014 to 8:00 at 11 March 2014.*

**Author's response:** Added in the revised manuscript.

**References:**

Aiken, A. C., Salcedo, D., Cubison, M. J., Huffman, J. A., DeCarlo, P. F., Ulbrich, I. M., Docherty, K. S., Sueper, D., Kimmel, J. R., Worsnop, D. R., Trimborn, A., Northway, M., Stone, E. A., Schauer, J. J., Volkamer, R. M., Fortner, E., de Foy, B., Wang, J., Laskin, A., Shutthanandan, V., Zheng, J., Zhang, R., Gaffney, J., Marley, N. A., Paredes-Miranda, G., Arnott, W. P., Molina, L. T., Sosa, G., and Jimenez, J. L.: Mexico City aerosol analysis during MILAGRO using high resolution aerosol mass spectrometry at the urban supersite (T0) – Part 1: Fine particle composition and organic source apportionment, Atmos. Chem. Phys., 9, 6633-6653, 2009.

Crippa, M., El Haddad, I., Slowik, J. G., DeCarlo, P. F., Mohr, C., Heringa, M. F., Chirico, R., Marchand, N., Sciare, J., Baltensperger, U., and Prévôt A. S. H.: Identification of marine and continental aerosol sources in Paris using high resolution aerosol mass spectrometry, J. Geophys. Res., 118, 1950–1963, 2013b.

Mohr, C., DeCarlo, P. F., Heringa, M. F., Chirico, R., Slowik, J. G., Richter, R., Reche, C., Alastuey, A., Querol, X., Seco, R., Peñuelas, J., Jiménez, J. L., Crippa, M., Zimmermann, R., Baltensperger, U. and Prévôt, A. S. H.: Identification and quantification of organic aerosol from cooking and other sources in Barcelona using aerosol mass spectrometer data, Atmos. Chem. Phys., 12, 1649–1665, 2012.

Setyan, A., Zhang, Q., Merkel, M., Knighton, W. B., Sun, Y., Song, C., Shilling, J. E., Onasch, T. B., Herndon, S. C., Worsnop, D. R., Fast, J. D., Zaveri, R. A., Berg, L. K., Wiedensohler, A., Flowers, B. A., Dubey, M. K., and Subramanian, R.: Characterization of submicron particles influenced by mixed biogenic and anthropogenic emissions using high-resolution aerosol mass spectrometry: results from CARES, Atmos. Chem. Phys., 12, 8131-8156, 2012.

Zhang, Q. Q., Jimenez, J.L., Canagaratna, M.R., Ulbrich, I.M., Ng, N.L., Worsnop, D.R., and Sun, Y.: Understanding atmospheric organic aerosols via factor analysis of aerosol mass spectrometry: a review, Analyt. Bioanalyt. Chem., 401, 3045-3067, 2011.

---

## Author Response (AR2)

**Author's response to Editor Report:**

*All reviewers and myself support publication of this article. However, I'm most concerned with the citation of Zotter et al. (in prep) for details of why the lower wavelength of 370 nm was not used in reply to Reviewers # 3 and 4. I do not think it is ever appropriate to make citations to unpublished work. Please remove Zotter et al. (in prep) from the references section and make sure the necessary details of Zotter et al. (in prep) are included in the SI section as suggested by Reviewer # 4. I don't think you can expect readers of your article to like references to unpublished work. Please make this change before I approve final publication.*

**Author's response:**

We thank the editor for pointing this out. Following his suggestion we have removed the citation in prep from the main text and have added in the SI a section containing all details on the choice of Ångström exponents and wavelengths used in this work. In the SI we now refer to the PhD. thesis of P. Zotter.

**Changes in text:**

*Page 8, Line 5:*

The Aethalometer measurements can be used to separate eBC from wood burning (eBC$_{wb}$) and from traffic (eBC$_{tr}$), by taking advantage of the spectral dependence of absorption, as described by the Ångström exponent (Ångström, 1929). Specifically, the enhanced absorption of wood burning particles in the ultraviolet and visible wavelengths region (370–520 nm) relative to that of traffic particles is used to separate the contributions of the two fractions. This method is described in detail in Sandradewi et al. (2008) and has been successfully applied at many locations across Europe (Favez et al., 2010; Herich et al., 2011; Sciare et al., 2011; Crilley et al., 2015). For a proper separation of the eBC fractions, the Aethalometer data was averaged to 30 minutes in order to increase the signal to noise. Thus, the obtained fractions eBC$_{wb}$ and eBC$_{tr}$ could only be used for the correlations with the external tracers, but their spatial distributions couldn't be explored. The absorption Ångström exponent was calculated using the absorption measured at 470 and 950 nm and Ångström exponents of 0.9 and 1.7 were used for traffic and wood burning, respectively. More details on the choice of the wavelengths and angstrom exponents are presented in the SI.

*New section in the SI:*

**SI for section 2.4.2. eBC source apportionment:**
The choice of the wavelengths and of the angstrom exponents used in this work are based on the findings in Zotter (2015), where radiocarbon ($^{14}$C) measurements of elemental carbon (EC) are combined with Aethalometer data to determine the Ångström exponents characteristic for wood burning ($\alpha_{wb}$) and traffic ($\alpha_{tr}$) emissions. The best α values were evaluated by fitting the source apportionment results of the Aethalometer (in particular $BC_{tr}/BC$) against the fossil fraction of EC ($EC_f/EC$) derived from $^{14}$C measurements. The best fitting $\alpha_{tr}$ = 0.9 and $\alpha_{wb}$ = 1.68 were obtained, when using the attenuation measured at 470 and 950 nm.
Other wavelength combinations were also tested but in all cases, especially when 370 nm was used, the residuals of the fit were worse. Moreover it is known that the 370 nm channel of the Aethalometer is more sensitive to artefacts, including response to light absorbing SOA and the adsorption of VOCs on the filter.

Zotter, P.: Sources of fossil and non-fossil atmospheric aerosols, Ph.D. thesis, Eidgenössische Technische Hochschule, ETH Zürich, Switzerland, 2015.

[revised manuscript text omitted]